# Decentralized Stochastic Nonconvex Optimization under the Relaxed Smoothness

## Abstract

This paper studies decentralized optimization problem $f(\mathbf{x}) = \frac{1}{m}\sum_{i=1}^{m} f_i(\mathbf{x})$, where each local function has the form of $f_i(\mathbf{x}) = \mathbb{E}\left[F(\mathbf{x};\boldsymbol{\xi}_i)\right]$ which is $(L_0, L_1)$-smooth but possibly nonconvex and the random variable $\boldsymbol{\xi}_i$ follows distribution $\mathcal{D}_i$. We propose a novel algorithm called decentralized normalized stochastic gradient descent (DNSGD), which can achieve an $\epsilon$-stationary point at each local agent. We present a new framework for analyzing decentralized first-order methods in the relaxed smooth setting, based on the Lyapunov function related to the product of the gradient norm and the consensus error. We show the upper bounds on the sample complexity of $\mathcal{O}(m^{-1}(L_f\sigma^2\Delta_f\epsilon^{-4} + \sigma^2\epsilon^{-2} + L_f^{-2}L_1^3\sigma^2\Delta_f\epsilon^{-1} + L_f^{-2}L_1^2\sigma^2))$ per agent and the communication complexity of $\tilde{\mathcal{O}}((L_f\epsilon^{-2} + L_1\epsilon^{-1})\gamma^{-1/2}\Delta_f)$, where $L_f = L_0 + L_1\zeta$, $\sigma^2$ is the variance of the stochastic gradient, $\Delta_f$ is the initial optimal function value gap, $\gamma$ is the spectral gap of the network, and $\zeta$ is the degree of the gradient dissimilarity. In the special case of $L_1 = 0$, the above results (nearly) match the lower bounds of decentralized stochastic nonconvex optimization under the standard smoothness. We also conduct numerical experiments to show the empirical superiority of our method.

## 1 Introduction

This paper considers the decentralized stochastic optimization over a connected network of $m$ nodes. Specifically, we study the minimization problem

$$\min_{\mathbf{x}\in\mathbb{R}^d} f(\mathbf{x}) = \frac{1}{m}\sum_{i=1}^{m} f_i(\mathbf{x}),$$

where $f_i : \mathbb{R}^d \to \mathbb{R}$ is the local function on $i$th agent and has the form of

$$f_i(\mathbf{x}) = \mathbb{E}\left[F_i(\mathbf{x};\boldsymbol{\xi}_i)\right].$$

We are interested in the setting that each $f_i$ is relaxed smooth (Zhang et al., 2020b) but possibly nonconvex and the random variable $\boldsymbol{\xi}_i$ follows distribution $\mathcal{D}_i$. The optimization theory and algorithms for relaxed smooth problem have received significant attention in recent years (Borodich & Kovalev, 2025; Chen et al., 2023; Chezhegov et al., 2025; Crawshaw & Liu, 2025; Gaash et al., 2025; Gorbunov et al., 2024; Jiang et al., 2025; Khirirat et al., 2024a;b; Koloskova et al., 2023; Li et al., 2023; 2024; Lobanov et al., 2024; Reisizadeh et al., 2025; Tovmasyan et al., 2025; Tyurin, 2025; Vankov et al., 2025; Xie et al., 2024; Yu et al., 2025; Zhang et al., 2020a), since the empirical studies suggest the relaxed smoothness can better characterize the training of neural network models than the classical smoothness (Cooper, 2022; Crawshaw et al., 2022; Zhang et al., 2020b). However, most existing works focus on the centralized setting, and how to well address the relaxed smooth optimization problem in the decentralized scenarios remains unclear.

The main challenge in decentralized optimization is that each agent can only access the information from itself or its neighbors, which leads to the consensus error among local variables (Nedić et al., 2018). For the standard smooth optimization, the gradient tracking technique successfully uses the gradient Lipschitz continuity to reduce the consensus error and guarantees the convergence of the algorithms (Nedic et al., 2017; Qu & Li, 2017). Additionally, the multi-consensus steps with Chebyshev acceleration (Arioli & Scott, 2014; Liu & Morse, 2011) can further improve the

communication complexity, leading to the optimal decentralized methods for both convex and nonconvex cases (Bai et al., 2024; Hendrikx et al., 2021; Li et al., 2022; Liu et al., 2024; Lu & De Sa, 2021; Luo et al., 2022; Metelev et al., 2024; Song et al., 2024; Xin et al., 2020; Ye et al., 2023; Yuan et al., 2022). For nonsmooth optimization, the recent works focus on finding the sub-optimal solutions or the approximate (generalized) stationary points of the Lipschitz continuous objective (Kovalev et al., 2024; Lan et al., 2020; Lin et al., 2024; Sahinoglu & Shahrampour, 2024), establishing efficient decentralized stochastic algorithms based on the boundedness of local (sub)gradients.

In this paper, we focus on decentralized optimization under the relaxed smoothness, which is more challenging than above classical settings since neither the local functions nor their gradients enjoy the Lipschitz continuity. We propose a novel algorithm called decentralized normalized stochastic gradient descent (DNSGD) to address above issues. Our method does not include the gradient clipping, avoiding the potential inconsistency in the update modes of different local variables arising from the consensus errors. We further construct a new Lyapunov function related to the product of the gradient norm and the consensus error, which successfully characterizes the convergence of our decentralized method in the relaxed smooth setting that allows local gradients to be neither bounded nor Lipschitz continuous. Moreover, we conduct experiments on the application of image classification to show the empirical advantage of our proposed method in practice.

## 2 PRELIMINARIES

We use the bold lowercase letter with the subscript to represent the local variable, e.g., vector $\mathbf{x}_i \in \mathbb{R}^d$ is a local variable on the $i$th agent. We use the bold uppercase letter to present the aggregated matrix consisting of local variables from all agents, e.g., $\mathbf{X} = [\mathbf{x}_1, \ldots, \mathbf{x}_m]^\top \in \mathbb{R}^{m \times d}$, where $\mathbf{x}_i \in \mathbb{R}^d$. We also denote the mean vector by the bold lowercase letter with bar, e.g., $\bar{\mathbf{x}} = \frac{1}{m}\mathbf{1}^\top \mathbf{X} \in \mathbb{R}^{1 \times d}$, where $\mathbf{1} = [1, \ldots, 1]^\top \in \mathbb{R}^m$. In addition, the notation $\|\cdot\|$ represents the Euclidean norm (or the Frobenius) norm for given vector (or matrix), and we use $\mathbf{I}$ to present the identity matrix. For the ease of presentation, we allow the input of the function be organized as either column vector or row vector.

We impose following common assumptions for nonconvex optimization under the relaxed smoothness.

**Assumption 1** (lower boundedness). *We suppose the objective $f$ is lower bounded, i.e., there exists some $f^* \in \mathbb{R}$ such that*

$$f^* = \inf_{\mathbf{x} \in \mathbb{R}^d} f(\mathbf{x}) > -\infty.$$

**Assumption 2** (relaxed smoothness). *We suppose each local function $f_i$ is $(L_0, L_1)$-smooth i.e., there exists some $L_0, L_1 \geq 0$ such that each local function $f_i$ satisfies*

$$\|\nabla f_i(\mathbf{x}) - \nabla f_i(\mathbf{y})\| \leq (L_0 + L_1 \|\nabla f_i(\mathbf{x})\|) \|\mathbf{x} - \mathbf{y}\| \tag{1}$$

*for all $\mathbf{x}, \mathbf{y} \in \mathbb{R}^d$ with $\|\mathbf{x} - \mathbf{y}\| \leq 1/L_1$, where we define $1/L_1 = +\infty$ if $L_1 = 0$.*

**Remark 1.** *The notion of relaxed smoothness was first introduced by Zhang et al. (2020b), which requires the second-order differentiability. Here, we consider the weaker condition (Assumption 2) characterized by the gradient, which is provided by Zhang et al. (2020a).*

We also introduce the assumption of the gradient dissimilarity to describe the heterogeneity of the local functions, which is widely used in distributed optimization (Assran et al., 2019; Gorbunov et al., 2021; Lian et al., 2017; Woodworth et al., 2020).

**Assumption 3** (bounded gradient dissimilarity). *We suppose there exists some $\zeta \geq 0$ such that the local gradient satisfies*

$$\|\nabla f_i(\mathbf{x}) - \nabla f(\mathbf{x})\| \leq \zeta$$

*for all $i \in [m]$ and $\mathbf{x} \in \mathbb{R}^d$.*

The following proposition shows that the objective $f$ is relaxed smooth if the local functions are relaxed smooth with bounded dissimilarity.

**Proposition 1.** *Under Assumptions 2 and 3, the objective function $f(\mathbf{x}) = \frac{1}{m}\sum_{i=1}^m f_i(\mathbf{x})$ is $(L_f, L_1)$-smooth with $L_f = L_0 + L_1\zeta$, i.e., we have*

$$\|\nabla f(\mathbf{x}) - \nabla f(\mathbf{y})\| \leq (L_f + L_1 \|\nabla f(\mathbf{x})\|) \|\mathbf{x} - \mathbf{y}\| \tag{2}$$

*for all $\mathbf{x}, \mathbf{y} \in \mathbb{R}^d$ with $\|\mathbf{x} - \mathbf{y}\| \leq 1/L_1$.*

---

**Algorithm 1** AccGossip($\mathbf{Y}^{(0)}, \mathbf{W}, K$)

---

1: $\mathbf{Y}^{(-1)} = \mathbf{Y}^{(0)}$

2: $\eta_y = (1 - \sqrt{1 - (\lambda_2(\mathbf{W}))^2})/(1 + \sqrt{1 - (\lambda_2(\mathbf{W}))^2})$

3: **for** $k = 0, 1, \ldots, K$ **do**

4: $\quad \mathbf{Y}^{(k+1)} = (1 + \eta_y)\mathbf{W}\mathbf{Y}^{(k)} - \eta_y\mathbf{Y}^{(k-1)}$

5: **end for**

6: **Output:** $\mathbf{Y}^{(K)}$

---

We are interested in the decentralize stochastic first-order methods, which can access the stochastic first-order oracle $\nabla F(\mathbf{x}; \boldsymbol{\xi}_i)$ that satisfies the following assumption.

**Assumption 4.** *For given $i \in [m]$ and $\mathbf{x} \in \mathbb{R}^d$, we suppose the $i$th agent can draw $\boldsymbol{\xi}_i \sim \mathcal{D}_i$ and access the stochastic first-order oracle $\nabla F_i(\mathbf{x}; \boldsymbol{\xi}_i)$ which satisfies*

$$\mathbb{E}[\nabla F_i(\mathbf{x}; \boldsymbol{\xi}_i)] = \nabla f_i(\mathbf{x}) \qquad and \qquad \mathbb{E}\left[\|\nabla F_i(\mathbf{x}; \boldsymbol{\xi}_i) - \nabla f_i(\mathbf{x})\|^2\right] \leq \sigma^2$$

*for some $\sigma > 0$.*

In decentralized setting, each agent can only exchange information with its neighbors in per communication round. One communication step in decentralized optimization typically can be presented by multiplication on the mixing matrix $\mathbf{W} \in \mathbb{R}^{m \times m}$, which satisfies the following standard assumption (Hendrikx et al., 2021; Scaman et al., 2017; Ye et al., 2023).

**Assumption 5.** *Let $\mathbf{W} \in \mathbb{R}^{m \times m}$ be the mixing matrix associated with the connected and undirected network. We suppose (a) the matrix $\mathbf{W}$ is symmetric with $w_{i,j} \geq 0$ for all $i, j \in [m]$, and $w_{i,j} \neq 0$ if and only if the $i$th agent and the $j$th agent are connected or $i = j$; (b) It holds $\mathbf{0} \preceq \mathbf{W} \preceq \mathbf{I}$, $\mathbf{W}^\top \mathbf{1} = \mathbf{W}\mathbf{1} = \mathbf{1}$, and $\mathrm{null}(\mathbf{I} - \mathbf{W}) = \mathrm{span}(\mathbf{1})$.*

Note that Assumption 5 implies the mixing matrix $\mathbf{W}$ is doubly stochastic, which indicates that multiplying by $\mathbf{W}$ on the aggregated matrix encourages all local variables converge to their average. We use $\lambda_2(\mathbf{W})$ to present the second largest eigenvalue of the mixing matrix $\mathbf{W}$. Under Assumption 5, we have $\lambda_2(\mathbf{W}) \in [0, 1)$, then we define the spectral gap of $\mathbf{W}$ as $\gamma = 1 - \lambda_2(\mathbf{W})$. Additionally, we present the multi-consensus steps with Chebyshev acceleration in Algorithm 1 (Arioli & Scott, 2014; Liu & Morse, 2011), which encourage all local variables to converge to the average as follows.

**Proposition 2** (Ye et al. (2023, Proposition 1)). *Under Assumption 5, the output of Algorithm 1 holds*

$$\frac{1}{m}\mathbf{1}^\top \mathbf{Y}^{(K)} = \bar{\mathbf{y}}^{(0)} \quad and \quad \|\mathbf{Y}^{(K)} - \mathbf{1}\bar{\mathbf{y}}^{(0)}\| \leq c_1 \big(1 - c_2\sqrt{1 - \lambda_2(\mathbf{W})}\big)^K \|\mathbf{Y}^{(0)} - \mathbf{1}\bar{\mathbf{y}}^{(0)}\|,$$

*where $\bar{\mathbf{y}}^{(0)} = \frac{1}{m}\mathbf{1}^\top \mathbf{Y}^{(0)}$, $c_1 = \sqrt{14}$, and $c_2 = 1 - 1/\sqrt{2}$.*

## 3 THE ALGORITHM AND MAIN RESULTS

We propose decentralized normalized stochastic gradient descent (DNSGD) in Algorithm 2, which combines the techniques of normalized gradient descent (Chen et al., 2023; Fang et al., 2018; Hazan et al., 2015; Li et al., 2024; Mandic, 2004), gradient tracking (Nedic et al., 2017; Qu & Li, 2017), and multi-consensus with Chebyshev acceleration (Hendrikx et al., 2021; Scaman et al., 2017; Ye et al., 2023). Here the matrices $\mathbf{X}^t$ and $\mathbf{V}^t$ consist of the local variables and local gradient estimators, i.e., $\mathbf{X}^t = [\mathbf{x}_1^t; \ldots; \mathbf{x}_m^t]^\top \in \mathbb{R}^{m \times d}$ and $\mathbf{V}^t = [\mathbf{v}_1^t; \ldots; \mathbf{v}_m^t]^\top \in \mathbb{R}^{m \times d}$.

It is worth noting that the relaxed smoothness condition (Assumption 2) does not guarantee that the local gradient is Lipschitz continuous or bounded. However, the existing analysis of decentralized methods mainly depends on the Lipschitz continuity or the boundedness of the local gradient (Bai et al., 2024; Hendrikx et al., 2021; Kovalev et al., 2024; Lan et al., 2020; Li et al., 2022; 2024; Lin et al., 2024; Liu et al., 2024; Lu & De Sa, 2021; Luo et al., 2022; Metelev et al., 2024; Sahinoglu & Shahrampour, 2024; Song et al., 2024; Xin et al., 2020; Ye et al., 2023; Yuan et al., 2022), which is not applicable to our setting.

---

**Algorithm 2** Decentralized Normalized Stochastic Gradient Descent (DNSGD)

---

1: **Input:** $\bar{\mathbf{x}}^0 \in \mathbb{R}^{1 \times d}$, $\mathbf{W} \in \mathbb{R}^{m \times m}$, $\eta > 0$, $\hat{K}, K, T, b \in \mathbb{N}$

2: $\mathbf{X}^0 = \mathbf{1}\bar{\mathbf{x}}^0$; independently sample $\boldsymbol{\xi}_{i,k}^0 \sim \mathcal{D}_i$ for all $k \in [b]$ at each agent $i$

3: $\mathbf{G}^0 = \left[ \dfrac{1}{b} \sum_{k=1}^{b} \nabla F_1(\bar{\mathbf{x}}^0; \boldsymbol{\xi}_{1,k}^0) \quad \cdots \quad \dfrac{1}{b} \sum_{k=1}^{b} \nabla F_m(\bar{\mathbf{x}}^0; \boldsymbol{\xi}_{m,k}^0) \right]^{\top}$

4: $\mathbf{V}^0 = \text{AccGossip}(\mathbf{G}^0, \mathbf{W}, \hat{K})$

5: **for** $t = 0, 1, \ldots, T-1$ **do**

6:     $\mathbf{U}^t = \left[ \dfrac{\mathbf{v}_1^t}{\|\mathbf{v}_1^t\|} \quad \cdots \quad \dfrac{\mathbf{v}_m^t}{\|\mathbf{v}_m^t\|} \right]^{\top}$

7:     $\mathbf{X}^{t+1} = \text{AccGossip}\left(\mathbf{X}^t - \eta \mathbf{U}^t, \mathbf{W}, K\right)$

8:     independently sample $\boldsymbol{\xi}_{i,k}^{t+1} \sim \mathcal{D}_i$ for all $k \in [b]$ at each agent $i$

9:     $\mathbf{G}^{t+1} = \left[ \dfrac{1}{b} \sum_{k=1}^{b} \nabla F_1(\mathbf{x}_1^{t+1}; \boldsymbol{\xi}_{1,k}^{t+1}) \quad \cdots \quad \dfrac{1}{b} \sum_{k=1}^{b} \nabla F_m(\mathbf{x}_m^{t+1}; \boldsymbol{\xi}_{m,k}^{t+1}) \right]^{\top}$

10:     $\mathbf{V}^{t+1} = \text{AccGossip}\left(\mathbf{V}^t + \mathbf{G}^{t+1} - \mathbf{G}^t, \mathbf{W}, K\right)$

11: **end for**

12: $\hat{\mathbf{x}}_i \sim \text{Unif}\{\mathbf{x}_i^0, \ldots, \mathbf{x}_i^{T-1}\}$ for each agent $i$

13: **Output:** $\hat{\mathbf{x}}_i$ for each agent $i$

---

To address above issues, we design the Lyapunov function

$$\Phi^t = f(\bar{\mathbf{x}}^t) + \frac{3\eta}{\sqrt{m}} \left(M_0 + M_1 \left\|\nabla f(\bar{\mathbf{x}}^t)\right\|\right) \left\|\mathbf{X}^t - \mathbf{1}\bar{\mathbf{x}}^t\right\| + \frac{2\eta}{\sqrt{m}} \left\|\mathbf{V}^t - \mathbf{1}\bar{\mathbf{v}}^t\right\|,$$

where $M_0 = \sqrt{2(L_0^2 + L_1^2\zeta^2)}$ and $M_1 = \sqrt{2}L_1$. In particular, our $\Phi_t$ includes the term

$$\left(M_0 + M_1 \left\|\nabla f(\bar{\mathbf{x}}^t)\right\|\right) \left\|\mathbf{X}^t - \mathbf{1}\bar{\mathbf{x}}^t\right\|$$

to align with the measurement unit on the right-hand side of inequality (2) for the relaxed smoothness of the objective. This design can well address the consensus errors $\|\mathbf{X}^t - \mathbf{1}\bar{\mathbf{x}}^t\|$ and $\|\mathbf{V}^t - \mathbf{1}\bar{\mathbf{v}}^t\|$ even if the local gradient is *non-Lipschitz* and *unbounded*.

In contrast, the existing analysis essentially only considers the linear combinations of the function value, gradient norm, and the consensus errors (Bai et al., 2024; Hendrikx et al., 2021; Kovalev et al., 2024; Lan et al., 2020; Li et al., 2022; 2024; Lin et al., 2024; Liu et al., 2024; Lu & De Sa, 2021; Luo et al., 2022; Metelev et al., 2024; Sahinoglu & Shahrampour, 2024; Song et al., 2024; Xin et al., 2020; Ye et al., 2023; Yuan et al., 2022), so that it does not work on the relaxed smoothness.

By using the Lyapunov function $\Phi^t$, we can show the main theoretical results of proposed DNSGD (Algorithm 2) as follows.

**Theorem 1.** *Under Assumptions 1–5, running DNSGD (Algorithm 2) with*

$$\eta = \min\left\{\frac{\epsilon}{4L_f + 1}, \frac{1}{2L_1}\right\}, \quad b \geq \max\left\{\frac{256(4L_f + 1)^2\sigma^2}{mL_f^2\epsilon^2}, \frac{1024L_1^2\sigma^2}{mL_f^2}\right\}$$

$$T \geq \max\left\{\frac{8(4L_f + 1)\Delta_\Phi}{\epsilon^2}, \frac{16L_1\Delta_\Phi}{\epsilon}\right\}, \quad \text{and} \quad K = \mathcal{O}\left(\frac{\log\left(\sqrt{m}(1 + \eta)\right)}{\sqrt{\gamma}}\right)$$

*guarantees* $\mathbb{E}\left[\|\nabla f(\hat{\mathbf{x}})\|\right] \leq \epsilon/2$, *where* $\hat{\mathbf{x}}$ *is uniformly sampled from* $\{\bar{\mathbf{x}}^0, \ldots, \bar{\mathbf{x}}^{T-1}\}$,

$$L_f = L_0 + L_1\zeta, \quad \text{and} \quad \Delta_\Phi = f(\bar{\mathbf{x}}^0) - \inf_{\mathbf{x} \in \mathbb{R}^d} f(\mathbf{x}) + \frac{2\eta}{\sqrt{m}}\mathbb{E}\left[\|\mathbf{V}^0 - \mathbf{1}\bar{\mathbf{v}}^0\|\right].$$

Note that the expected $(\epsilon/2)$-stationary point $\hat{\mathbf{x}}$ in Theorem 1 is based on the mean vectors $\{\bar{\mathbf{x}}^t\}_{t=0}^{T-1}$, which are unavailable in decentralized setting. In addition, the expression of $\Delta_\Phi$ indicates the initial consensus error $\|\mathbf{V}^0 - \mathbf{1}\bar{\mathbf{v}}^0\|$ may heavily affect the iteration number $T$ if it is significantly larger than $\Delta_f = f(\bar{\mathbf{x}}^0) - \inf_{\mathbf{x} \in \mathbb{R}^d} f(\mathbf{x})$. Fortunately, we can address above issues by appropriate setting of $\hat{K}$ in line 4 to reduce the magnitude of $\|\mathbf{V}^0 - \mathbf{1}\bar{\mathbf{v}}^0\|$ to achieve $\Delta_\Phi = \Theta(\Delta_f)$.

Finally, we achieve the refined upper complexity bounds as follows.

**Corollary 1.** *Following the setting of Theorem 1, we further let $\hat{K} = \tilde{\mathcal{O}}(1/\sqrt{\gamma})$, then it holds that $\Delta_\Phi = \Theta(\Delta_f)$ and the output of DNSGD (Algorithm 2) guarantees $\mathbb{E}\left[\|\nabla f(\hat{\mathbf{x}}_i)\|\right] \le \epsilon$ for each agent $i \in [m]$, where $\hat{\mathbf{x}}_i$ is uniformly sampled from $\{\mathbf{x}_i^0, \ldots, \mathbf{x}_i^{T-1}\}$ generated by the $i$th agent. Additionally, each agent requires the sample complexity of*

$$Tb = \mathcal{O}\left(\frac{1}{m}\left(\frac{L_f \sigma^2 \Delta_f}{\epsilon^4} + \frac{\sigma^2}{\epsilon^2} + \frac{L_1^3 \sigma^2 \Delta_f}{L_f^2 \epsilon} + \frac{L_1^2 \sigma^2}{L_f^2}\right)\right) \tag{3}$$

*and the algorithm requires the communication complexity of*

$$TK + \hat{K} = \tilde{\mathcal{O}}\left(\frac{L_f \Delta_f}{\sqrt{\gamma}\epsilon^2} + \frac{L_1 \Delta_f}{\sqrt{\gamma}\epsilon}\right),$$

*where $L_f = L_0 + L_1 \zeta$, $\Delta_f = f(\bar{\mathbf{x}}^0) - \inf_{\mathbf{x} \in \mathbb{R}^d} f(\mathbf{x})$, and the notation $\tilde{\mathcal{O}}(\cdot)$ hides the log factors.*

The term $1/m$ in the sample complexity bound (3) implies our method achieves the linear speedup. Although expression (3) looks somewhat complicated, it is easy to understand since we can verify our bounds match or surpass the best-known results in several special cases:

- In the case of $L_1 = 0$, the local functions reduce to the setting of standard $L_0$-smooth and it leads the objective $f$ is also $L_0$-smooth without Assumption 3 and $L_f = L_0$. Then Corollary 1 indicates the sample complexity and the communication complexity of

$$Tb = \mathcal{O}\left(\frac{1}{m}\left(\frac{L_0 \sigma^2 \Delta_f}{\epsilon^4} + \frac{\sigma^2}{\epsilon^2}\right)\right) \qquad \text{and} \qquad TK + \hat{K} = \tilde{\mathcal{O}}\left(\frac{L_0 \Delta_f}{\sqrt{\gamma}\epsilon^2}\right),$$

respectively. This matches the (near-)optimal results for decentralized stochastic nonconvex optimization under the standard smoothness (Lu & De Sa, 2021, Corollary 1).

- In the case of $m = 1$, the problem reduces to the relaxed smooth optimization on single machine. This leads to $\zeta = 0$ and $L_f = L_0$ since there is no heterogeneity. Then Theorem 1 indicates the sample complexity of

$$Tb = \mathcal{O}\left(\frac{L_0 \sigma^2 \Delta_f}{\epsilon^4} + \frac{\sigma^2}{\epsilon^2} + \frac{L_1^3 \sigma^2 \Delta_f}{L_0^2 \epsilon} + \frac{L_1^2 \sigma^2}{L_0^2}\right), \tag{4}$$

which is tighter than the complexity of gradient clipping method in the non-distributed case (Zhang et al., 2020a). Recall that Theorem 3.2 of Zhang et al. (2020a) claims the sample complexity of $\mathcal{O}(L_0 \sigma^2 \Delta_f \epsilon^{-4} + L_0^{-3} L_1^4 \sigma^2 \Delta_f)$. It is worth noting that term $L_0^{-3} L_1^4 \sigma^2 \Delta_f$ dominates their bound only when $\epsilon = \Omega(L_0/L_1)$, and it is larger than the term $L_0^{-2} L_1^3 \sigma^2 \Delta_f \epsilon^{-1}$ in our (4) in this case. In addition, Zhang et al. (2020a) assume the algorithm can access the exact gradient at the initial point, while their complexity bound does not include the cost of achieving the the exact gradient at the initial point. In contrast, our analysis considers the access of the initial gradient estimator by stochastic first-order oracle, leading to the terms $\sigma^2 \epsilon^{-2}$ and $L_1^2 L_0^{-2} \sigma^2$ in our upper bound (4). Recently, Liu & Zhou (2024) study the more general stochastic gradient estimator with heavy-tailed noise, while applying their result to our setting leads to the term $\sigma^3 \epsilon^{-3}$ in the upper bound on the sample complexity (Liu & Zhou, 2024, Theorem 3.2 with $p = 2$).

- Following the case of $m = 1$, we further consider the deterministic case, i.e., replace the mini-batch stochastic gradient in Algorithm 2 with the exact gradient and omit the communication steps. Then Theorem 1 indicates the iteration (gradient) complexity of

$$T = \mathcal{O}\left(\frac{L_0 \Delta_f}{\epsilon^2} + \frac{L_1 \Delta_f}{\epsilon}\right),$$

matching the best-known result for solving relaxed smooth problem by exact gradient method on the single machine (Vankov et al., 2025, Theorem 3.1).

## 4 THE COMPLEXITY ANALYSIS

In this section, we briefly sketch the derivations of our main results. The details for all proofs are presented in the appendices. We first present the descent lemma with respect to the mean vectors $\bar{\mathbf{x}}^{t+1} = \frac{1}{m}\sum_{i=1}^{m}\mathbf{x}_i^{t+1}$ and $\bar{\mathbf{x}}^t = \frac{1}{m}\sum_{i=1}^{m}\mathbf{x}_i^t$ as follows.

**Lemma 1.** *Under the setting of Theorem 1, we have*

$$\mathbb{E}[f(\bar{\mathbf{x}}^{t+1})] \leq f(\bar{\mathbf{x}}^t) - \eta\left(1 - \frac{\eta L_1}{2}\right)\left\|\nabla f(\bar{\mathbf{x}}^t)\right\| + \frac{2\sqrt{2}\eta}{\sqrt{m}}\left(M_0 + M_1\left\|\nabla f(\bar{\mathbf{x}}^t)\right\|\right)\left\|\mathbf{X}^t - \mathbf{1}\bar{\mathbf{x}}^t\right\|$$

$$+ \frac{\eta}{\sqrt{m}}\left\|\mathbf{V}^t - \mathbf{1}\bar{\mathbf{v}}^t\right\| + \frac{2\eta\sigma}{\sqrt{mb}} + \frac{\eta^2 L_f}{2}$$

*for given $\mathbf{X}^t$ and $\mathbf{V}^t$.*

We define $\rho = c_1(1 - c_2\sqrt{1 - \lambda_2(\mathbf{W})})^K$ with $c_1 = \sqrt{14}$ and $c_2 = 1 - 1/\sqrt{2}$. Proposition 2 implies we can guarantee $\rho$ be sufficient small by taking appropriate communication rounds number $K$. We then establish the recursions for the consensus errors as follows.

**Lemma 2.** *For Algorithm 2, we have*

$$\left\|\mathbf{X}^{t+1} - \mathbf{1}\bar{\mathbf{x}}^{t+1}\right\| \leq \rho\left(\left\|\mathbf{X}^t - \mathbf{1}\bar{\mathbf{x}}^t\right\| + m\eta\right) \qquad and \qquad \left\|\mathbf{X}^t - \mathbf{1}\bar{\mathbf{x}}^t\right\| \leq \frac{\rho m\eta}{1 - \rho}.$$

**Lemma 3.** *For Algorithm 2 with $K \geq \left((2+\sqrt{2})/\sqrt{\gamma}\right)\log\sqrt{14}(1 + m\eta L_1)$, we have*

$$\mathbb{E}\left[\left\|\mathbf{V}^{t+1} - \mathbf{1}\bar{\mathbf{v}}^{t+1}\right\|\right]$$

$$\leq \rho\left(\left\|\mathbf{V}^t - \mathbf{1}\bar{\mathbf{v}}^t\right\| + \left(M_2 + M_3\left\|\nabla f(\bar{\mathbf{x}}^t)\right\|\right)\left\|\mathbf{X}^t - \mathbf{1}\bar{\mathbf{x}}^t\right\| + \left(M_2 + M_3\left\|\nabla f(\bar{\mathbf{x}}^t)\right\|\right)m\eta + \frac{2\sqrt{m}\sigma}{\sqrt{b}}\right),$$

*for given $\mathbf{X}^t$ and $\mathbf{V}^t$, where $M_2 = (\rho+1)L_f + \rho L_f L_1\eta$ and $M_3 = \rho L_1(L_1\eta + 1) + L_1$.*

Applying Lemmas 1, 2, and 3, we can characterize the decease of the Lyapunov function as

$$\mathbb{E}[\Phi^{t+1}] \leq \Phi^t - \frac{5\eta}{8}\left\|\nabla f(\bar{\mathbf{x}}^t)\right\| + \frac{3\eta^2 L_f}{4}. \tag{5}$$

Summing equation (5) over $t = 0, \ldots, T-1$, we can achieve our main result in Theorem 1 that guarantees $\mathbb{E}\left[\left\|\nabla f(\hat{\mathbf{x}})\right\|\right] \leq \epsilon/2$, where $\hat{\mathbf{x}}$ is uniformly sampled from $\{\bar{\mathbf{x}}^1, \ldots, \bar{\mathbf{x}}^m\}$.

In Theorem 1, the setting of iterations number $T \geq \max\left\{8(4L_f + 1)\Delta_\Phi\epsilon^{-2}, 16L_1\Delta_\Phi\epsilon^{-1}\right\}$ depends on the term $\Delta_\Phi = f(\bar{\mathbf{x}}^0) - \inf_{\mathbf{x}\in\mathbb{R}^d} f(\mathbf{x}) + (2\eta/\sqrt{m})\mathbb{E}\left[\left\|\mathbf{V}^0 - \mathbf{1}\bar{\mathbf{v}}^0\right\|\right]$, which may be dominated by the expected consensus error $\mathbb{E}\left[\left\|\mathbf{V}^0 - \mathbf{1}\bar{\mathbf{v}}^0\right\|\right]$. However, the results in the single machine scenario only depend on $\Delta_f = f(\bar{\mathbf{x}}^0) - \inf_{\mathbf{x}\in\mathbb{R}^d} f(\mathbf{x})$. To fill this gap, we set the initial communication rounds to be $\hat{K} = \tilde{\mathcal{O}}(1/\sqrt{\gamma})$ for line 4 of Algorithm 2, then Proposition 2 guarantees the consensus error satisfies $(2\eta/\sqrt{m})\mathbb{E}\left[\left\|\mathbf{V}^0 - \mathbf{1}\bar{\mathbf{v}}^0\right\|\right] \leq \mathcal{O}(\Delta_f)$ and achieves $\Delta_\Phi \leq \mathcal{O}(\Delta_f)$. Therefore, the upper bound of $T$ to guarantee $\mathbb{E}\left[\left\|\nabla f(\hat{\mathbf{x}})\right\|\right] \leq \epsilon/2$ can be improved to $\mathcal{O}(L_f\Delta_f\epsilon^{-2} + L_1\Delta_f\epsilon^{-1})$.

For achieving an $\epsilon$-stationary point on each node, we consider the gradient at $\hat{\mathbf{x}}_i$ as follows

$$\mathbb{E}[\left\|\nabla f(\hat{\mathbf{x}}_i)\right\|] = \mathbb{E}\left[\frac{1}{T}\sum_{t=0}^{T-1}\left\|\nabla f(\mathbf{x}_i^t)\right\|\right] \leq \mathbb{E}\left[\underbrace{\frac{1}{T}\sum_{t=0}^{T-1}\left\|\nabla f(\mathbf{x}_i^t) - \nabla f(\bar{\mathbf{x}}^t)\right\|}_{A_1} + \underbrace{\frac{1}{T}\sum_{t=0}^{T-1}\left\|\nabla f(\bar{\mathbf{x}}^t)\right\|}_{A_2}\right].$$

For term $A_1$, Assumption 2 implies

$$\mathbb{E}[A_1] \leq \mathbb{E}\left[\frac{1}{T}\sum_{t=0}^{T-1}\left(L_f + L_1\left\|\nabla f(\bar{\mathbf{x}}^t)\right\|\right)\left\|\mathbf{X}^t - \mathbf{1}\bar{\mathbf{x}}^t\right\|\right] \leq \mathbb{E}\left[\frac{1}{T}\sum_{t=0}^{T-1}\frac{\rho m\eta(L_f + L_1 TB)}{1 - \rho}\right],$$

where the last step is based on Lemma 8. Recall that Theorem 1 shows $\mathbb{E}[A_2] = \mathbb{E}[\left\|\nabla f(\hat{\mathbf{x}})\right\|] \leq \epsilon/2$ and we can set appropriate $K = \tilde{\mathcal{O}}(\sqrt{1/\gamma})$ to make $\rho$ be sufficient small to guarantee $\mathbb{E}[A_1] \leq \epsilon/2$. Therefore, we achieve the desired $\epsilon$-stationary point $\hat{\mathbf{x}}_i$ at each agent $i$.

## 5 DISCUSSION AND RELATED WORK

The design of our DNSGD (Algorithm 2) is based on normalized (stochastic) gradient descent, which is extensively studied in centralized scenario (Chen et al., 2023; Cutkosky & Mehta, 2020; Fang et al., 2018; Hazan et al., 2015; Mandic, 2004). In a related work, Li et al. (2024) studied normalized stochastic gradient descent in decentralized setting, while their analysis is only limited to the setting of standard smoothness. In a very recent work, Jiang et al. (2025, Algorithm 2 and Theorem 4) claims a first-order method for decentralized optimization under the relaxed smoothness, while each of their iteration requires every node to perform the step of gradient clipping which depends on accessing the mean vector $\bar{\mathbf{x}}_i^t$ and the corresponding exact local gradient $\nabla f_i(\bar{\mathbf{x}}^t)$. Recall that the decentralized method typically only allows each node directly communicates with its neighbors (Assumption 5) and the stochastic first-order oracle can only access the unbiased gradient estimator with bounded variance (Assumption 4). Therefore, we argue that their algorithm is not truly decentralized or stochastic. In contrast, our method only requires each node to iterate with its local stochastic gradient (line 9 of Algorithm 2), which is fully decentralized and stochastic.

The convergence analysis for our DNSGD (Algorithm 2) depends on the assumption of bounded gradient dissimilarity (Assumption 3), which is commonly used in distributed optimization (Assran et al., 2019; Gorbunov et al., 2021; Lian et al., 2017; Woodworth et al., 2020). Specifically, we use this assumption to establish the inequality $\left\| \nabla f(\bar{\mathbf{x}}^{t+1}) \right\| \leq (L_0 + L_1 \zeta) \eta + (1 + L_1 \eta) \left\| \nabla f(\bar{\mathbf{x}}^t) \right\|$, which connects the gradients at the points $\bar{\mathbf{x}}^t$ and $\bar{\mathbf{x}}^{t+1}$ (see Lemma 9 in Appendix C). We also combine Assumption 3 with the $(L_0, L_1)$-smoothness of local functions (Assumption 2) to guarantee the global objective $f = \frac{1}{m} \sum_{i=1}^{m} f_i$ is $(L_0 + L_1 \zeta, L_1)$-smooth (see the proof in Appendix B). However, how to establish the convergence for decentralized relaxed smooth optimization without bounded gradient dissimilarity (Assumption 3) is still an open problem. It is worth noting that simply assuming that each local function $f_i$ is $(L_0, L_1)$-smooth even cannot guarantee that the global function $f = \frac{1}{m} \sum_{i=1}^{m} f_i$ is also $(L_0, L_1)$-smooth, which is different with the standard smooth case. We illustrate this point with the following example.

**Proposition 3.** *We consider functions*

$$f_1(x) = \exp(Lx), \quad f_2(x) = \exp(-Lx), \quad \text{and} \quad f(x) = \frac{f_1(x) + f_2(x)}{2}$$

*for some $L > 0$ and let $L_1 = L/\log 2$, then both functions $f_1$ and $f_2$ is $(0, L_1)$-smooth while the function $f$ is not $(0, L_1)$-smooth.*

Recall that the discussion after Corollary 1 shows that the sample complexity of our method matches the lower bound in the special case of $L_1 = 0$ (Lu & De Sa, 2021). This implies that the $\mathcal{O}(\epsilon^{-4})$ dependence on accuracy we obtained cannot be improved in our setting. If we target to achieve the sharper $\mathcal{O}(\epsilon^{-3})$ dependence in the sample complexity, the assumption of expected relaxed smoothness is necessary (Chen et al., 2023; Reisizadeh et al., 2025), e.g., strengthen condition (1) in Assumption 2 to $\mathbb{E}[\|\nabla F_i(\mathbf{x}; \boldsymbol{\xi}_i) - \nabla F_i(\mathbf{y}; \boldsymbol{\xi}_i)\|^2] \leq (L_0 + L_1 \|\nabla f_i(\mathbf{x})\|)^2 \|\mathbf{x} - \mathbf{y}\|^2$, then applying the variance-reduced gradient estimators (Chen et al., 2023; Fang et al., 2018; Reisizadeh et al., 2025) at local agents is possible to achieve the sample complexity depends on $\mathcal{O}(\epsilon^{-3})$.

## 6 EXPERIMENTS

In this section, we empirically compare our decentralized normalized stochastic gradient descent (DNSGD) with baseline methods, including decentralized stochastic gradient descent (D-SGD) (Lian et al., 2017), decentralized stochastic gradient descent with gradient tracking (D-SGT) (Nedic & Ozdaglar, 2009; Qu & Li, 2017), and decentralized normalized averaged stochastic approximation (D-NASA) (Li et al., 2024). The experiments are performed on a cluster with 8 virtual GPUs, 8 virtual CPUs, and 96 GB memory.

We present numerical results on the applications of image classification for MNIST and Fashion-MNIST datasets. The MNIST dataset is a well-known collection of handwritten digits (10 classes), containing $70,000$ images, with $60,000$ for training and $10,000$ for testing (LeCun et al., 2002). The Fashion-MNIST dataset serves as a more challenging variant, consisting of $70,000$ images of fashion items with 10 classes, such as trouser and dress (Xiao et al., 2017). Each example of MNIST or Fashion-MNIST is a $28 \times 28$ grayscale image.

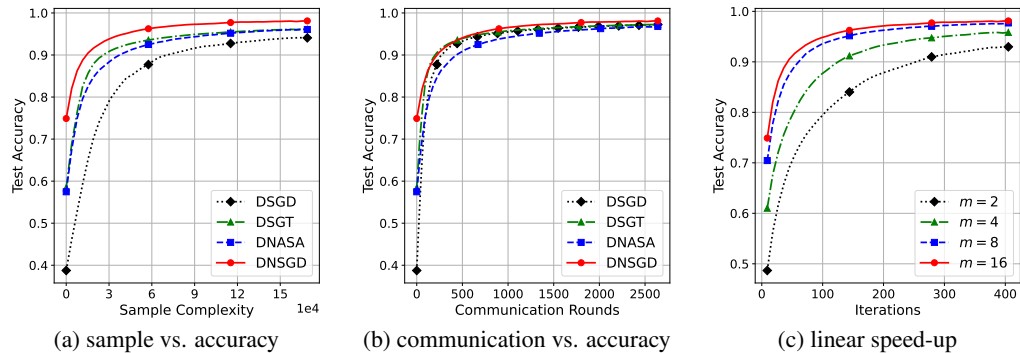

(a) sample vs. accuracy      (b) communication vs. accuracy      (c) linear speed-up

Figure 1: We present numerical results on the MNIST dataset over the Erdös-Rényi network. The subfigures (a) and (b) present the sample complexity per agent and the communication rounds against the average test accuracy on the network with $m = 16$ agents. The subfigure (c) verifies the linear speedup effect of our DNSGD on the network with $m = 2, 4, 8, 16$ agents.

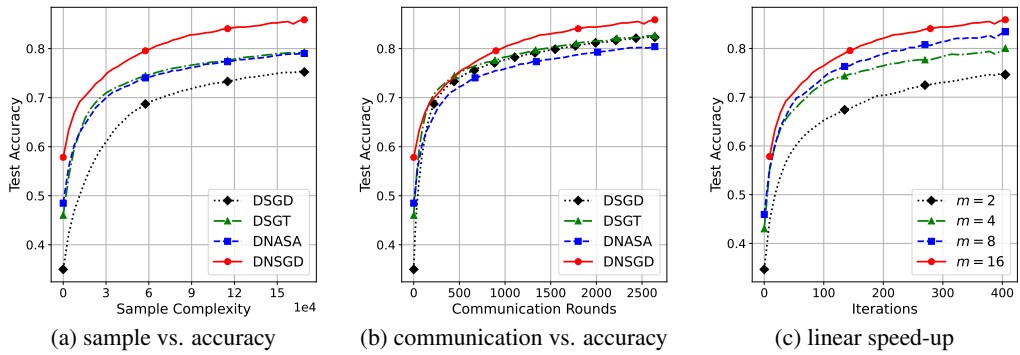

(a) sample vs. accuracy      (b) communication vs. accuracy      (c) linear speed-up

Figure 2: We present numerical results on the Fashion-MNIST dataset over the Erdös-Rényi network. The subfigures (a) and (b) present the sample complexity per agent and the communication rounds against the average test accuracy on the network with $m = 16$ agents. The subfigure (c) verifies the linear speedup effect of our DNSGD on the network with $m = 2, 4, 8, 16$ agents.

Our model employs a four-layer neural network, consisting of two convolutional layers and two fully connected layers. The first convolutional layer processes the single-channel input image with 32 kernels of size $3 \times 3$, followed by ReLU activation and $2 \times 2$ max-pooling. The second convolutional layer takes the 32-channel feature maps from the first layer, applies another 32 kernels of size $3 \times 3$, and is followed by ReLU activation and $2 \times 2$ max-pooling. To enhance generalization, we apply dropout with a rate of $0.25$ after the convolutional layers. The resulting feature maps are then flattened and passed through a fully connected layer with $128$ units and ReLU activation, followed by a second dropout layer with a rate of $0.5$. Finally, we use a fully connected output layer to generate scores for the 10 classes.

We perform numerical comparisons on a distributed system with 16 agents, where the data is evenly distributed among them. In our experiments, we employ two network topologies, i.e., the Erdös-Rényi graph (Erdős & Rényi, 1959) with the connectivity probability of $p = 0.6$ and the ring graph. For all algorithms, we set batch size to $b = 250$. The step size of D-SGD, D-SGT and DNSGD are chosen from $\{0.0001, 0.0005, 0.001, 0.005, 0.01, 0.05, 0.1, 0.5\}$. The learning rate of the D-NASA algorithm is given by $\eta_t = m^{1/4} t^{3/4}$ which follows the theory and implementation of Li et al. (2024), where $t$ is the iteration counter and $m$ is the number of agents. For our DNSGD, we set the iterations number of Chebyshev acceleration (Algorithm 1) to $\hat{K} = K = 2$. We present the numerical results on the Erdös-Rényi graph in Figures 1 and 2, and the results on the ring graph are shown in Figures 3 and 4. We evaluation the performance by showing the sample complexity and the communication rounds against the average test accuracy, i.e., the mean of the test accuracies achieved by the models from all agents. We can observe that DNSGD outperforms all baseline methods. The linear speed-up of our method is verified by comparing the results on the networks with $m = 2, 4, 8, 16$ agents. Figures 1(c), 2(c), 3(c), and 4(c) show that the more agents do lead to the faster convergence.

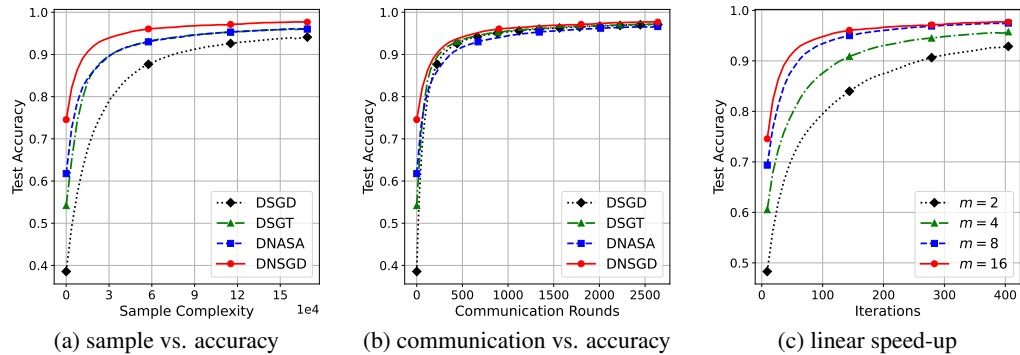

(a) sample vs. accuracy      (b) communication vs. accuracy      (c) linear speed-up

Figure 3: We present numerical results on the MNIST dataset over the ring network. The subfigures (a) and (b) present the sample complexity per agent and the communication rounds against the average test accuracy on the network with $m = 16$ agents. The subfigure (c) verifies the linear speedup effect of our DNSGD on the network with $m = 2, 4, 8, 16$ agents.

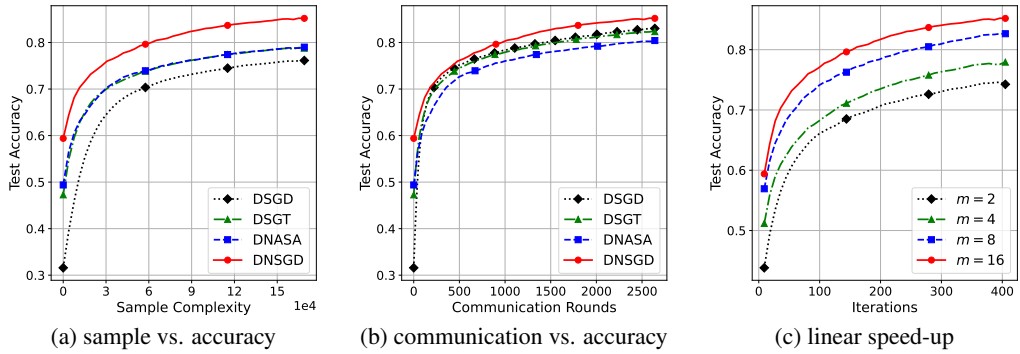

(a) sample vs. accuracy      (b) communication vs. accuracy      (c) linear speed-up

Figure 4: We present numerical results on the Fashion-MNIST dataset over the ring network. The subfigures (a) and (b) present the sample complexity per agent and the communication rounds against the average test accuracy on the network with $m = 16$ agents. The subfigure (c) verifies the linear speedup effect of our DNSGD on the network with $m = 2, 4, 8, 16$ agents.

## 7 CONCLUSION

This paper proposes decentralized normalized stochastic gradient descent for nonconvex optimization under the relaxed smoothness. We provide theoretical analysis to show our method can achieve the $\epsilon$-stationary point on each agent. In particular, we design a new Lyapunov function including the product of the gradient norm and the consensus error for convergence analysis, which successfully address the non-Lipschitz and unbounded local gradients in relaxed smooth decentralized optimization. The empirical studies also show that our method performs better than baselines. In future work, we would like to establish decentralized methods for convex optimization under the relaxed smoothness (Chezhegov et al., 2025; Gaash et al., 2025; Gorbunov et al., 2024; Koloskova et al., 2023; Lobanov et al., 2024; Vankov et al., 2025) and study the possibility to use the momentum to improve the performance of the algorithm in practice Cutkosky & Mehta (2020). It is also interesting to extend our results to the more general settings such as the symmetric relaxed smoothness (Chen et al., 2023) and the stochastic gradient estimator with heavy-tailed noise (Liu & Zhou, 2024).

## REPRODUCIBILITY STATEMENT

We provide the complete proofs for all theoretical claims in the appendices. We include the source code in the supplementary material for the reproducibility. The datasets MNIST[1] and Fashion-MNIST[2] used in our experiments are publicly available.

---

[1] http://yann.lecun.com/exdb/mnist/
[2] http://github.com/zalandoresearch/fashion-mnist/

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

## A  SOME TECHNICAL LEMMAS

We first provide some technical lemmas for later proofs.

**Lemma 4** (Ye et al. (2023, Lemma 12))**.** *For given matrix* $\mathbf{G} = [\mathbf{g}_1; \ldots; \mathbf{g}_m] \in \mathbb{R}^{m \times d}$, *we have* $\|\mathbf{G} - \mathbf{1}\bar{\mathbf{g}}\|^2 \leq \|\mathbf{G}\|^2$, *where* $\bar{\mathbf{g}} = \frac{1}{m} \sum_{i=1}^{m} \mathbf{g}_i$.

**Lemma 5** (Li et al. (2024, Lemma B.5))**.** *For given vectors* $\mathbf{v}_1, \ldots, \mathbf{v}_m \in \mathbb{R}^{1 \times d}$ *and* $\bar{\mathbf{v}} = \frac{1}{m} \sum_{i=1}^{m} \mathbf{v}_i$, *we have*

$$\left\| \frac{1}{m} \sum_{i=1}^{m} \frac{\mathbf{v}_i}{\|\mathbf{v}_i\|} - \frac{\bar{\mathbf{v}}}{\|\bar{\mathbf{v}}\|} \right\| \leq \frac{1}{m} \sum_{i=1}^{m} \frac{\|\bar{\mathbf{v}} - \mathbf{v}_i\|}{\|\bar{\mathbf{v}}\|}. \tag{6}$$

*Proof.* This result is presented in the analysis Li et al. (2024, Lemma B.5). Here, we provide the detailed proof for completeness. We have

$$\left\| \frac{1}{m} \sum_{i=1}^{m} \frac{\mathbf{v}_i}{\|\mathbf{v}_i\|} - \frac{\bar{\mathbf{v}}}{\|\bar{\mathbf{v}}\|} \right\| = \left\| \frac{1}{m} \sum_{i=1}^{m} \frac{\mathbf{v}_i}{\|\mathbf{v}_i\|} - \frac{1}{m} \sum_{i=1}^{m} \frac{\mathbf{v}_i}{\|\bar{\mathbf{v}}\|} \right\|$$

$$= \left\| \frac{1}{m} \sum_{i=1}^{m} \frac{(\|\bar{\mathbf{v}}\| - \|\mathbf{v}_i\|)\mathbf{v}_i}{\|\mathbf{v}_i\| \|\bar{\mathbf{v}}\|} \right\| \leq \frac{1}{m} \sum_{i=1}^{m} \left\| \frac{(\|\bar{\mathbf{v}}\| - \|\mathbf{v}_i\|)\mathbf{v}_i}{\|\mathbf{v}_i\| \|\bar{\mathbf{v}}\|} \right\|$$

$$\leq \frac{1}{m} \sum_{i=1}^{m} \frac{|\|\bar{\mathbf{v}}\| - \|\bar{\mathbf{v}}_i\|| \, \|\mathbf{v}_i\|}{\|\mathbf{v}_i\| \|\bar{\mathbf{v}}\|} = \frac{1}{m} \sum_{i=1}^{m} \frac{|\|\bar{\mathbf{v}}\| - \|\mathbf{v}_i\||}{\|\bar{\mathbf{v}}\|}$$

$$\leq \frac{1}{m} \sum_{i=1}^{m} \frac{\|\bar{\mathbf{v}} - \mathbf{v}_i\|}{\|\bar{\mathbf{v}}\|},$$

where the first equality is based on the fact $\bar{\mathbf{v}} = \frac{1}{m} \sum_{i=1}^{m} \mathbf{v}_i$; the first inequality is based on vector AM-QM inequality; the second inequality is based on Cauchy–Schwarz inequality; the last step is based on the triangle inequality. $\square$

**Lemma 6.** *The update of Algorithm 2 implies* $\bar{\mathbf{v}}^t = \bar{\mathbf{g}}^t$.

*Proof.* This result can be obtained by following Luo et al. (2022, Lemma 2). Here, we provide the detailed proof for completeness. The update in line 4 and Proposition 2 imply $\bar{\mathbf{v}}^0 = \bar{\mathbf{g}}^0$. Suppose it holds $\bar{\mathbf{v}}^t = \bar{\mathbf{g}}^t$, then the update in line 10 means

$$\bar{\mathbf{v}}^{t+1} = \frac{1}{m} \sum_{i=1}^{m} (\mathbf{v}_i^t + \mathbf{g}_i^{t+1} - \mathbf{g}_i^t) = \bar{\mathbf{v}}^t + \bar{\mathbf{g}}^{t+1} - \bar{\mathbf{g}}^t = \bar{\mathbf{g}}^{t+1}.$$

Therefore, we finish the proof by induction. $\square$

**Lemma 7** (Zhang et al. (2020a, Appendix A.1))**.** *Under Assumptions 2 and 3, we have*

$$f(\mathbf{y}) \leq f(\mathbf{x}) + \langle \nabla f(\mathbf{x}), \mathbf{y} - \mathbf{x} \rangle + \frac{L_f + L_1 \|\nabla f(\mathbf{x})\|}{2} \|\mathbf{y} - \mathbf{x}\|^2 \tag{7}$$

*for all* $\mathbf{x}, \mathbf{y} \in \mathbb{R}^d$ *with* $\|\mathbf{y} - \mathbf{x}\| \leq 1/L_1$.

*Proof.* Proposition 1 means the objective $f$ is $(L_f, L_1)$-smooth, then applying equation (29) of Zhang et al. (2020a, Appendix A.1) directly achieves equation (7). $\square$

## B    THE PROOF OF PROPOSITION 1

*Proof.* For all $\mathbf{x}, \mathbf{y} \in \mathbb{R}^d$ with $\|\mathbf{x} - \mathbf{y}\| \leq 1/L_1$, we have

$$
\|\nabla f(\mathbf{x}) - \nabla f(\mathbf{y})\|^2
$$

$$
= \left\| \frac{1}{m} \sum_{i=1}^{m} \left( \nabla f_i(\mathbf{x}) - \nabla f_i(\mathbf{y}) \right) \right\|^2
$$

$$
\leq \frac{1}{m} \sum_{i=1}^{m} \|\nabla f_i(\mathbf{x}) - \nabla f_i(\mathbf{y})\|^2
$$

$$
\leq \frac{1}{m} \sum_{i=1}^{m} (L_0 + L_1 \|\nabla f_i(\mathbf{x})\|)^2 \|\mathbf{x} - \mathbf{y}\|^2
$$

$$
\leq (L_0 + L_1 (\|\nabla f_i(\mathbf{x}) - \nabla f(\mathbf{x})\| + \|\nabla f(\mathbf{x})\|))^2 \|\mathbf{x} - \mathbf{y}\|^2
$$

$$
= (L_0 + L_1 \zeta + L_1 \|\nabla f(\mathbf{x})\|)^2 \|\mathbf{x} - \mathbf{y}\|^2 ,
$$

where the first inequality is based on the Young's inequality; the second inequality is based on Assumption 2; the third inequality is based on Assumption 3. Therefore, the function $f$ is $(L_f, L_1)$-smooth with $L_f = L_0 + L_1 \zeta$.                   □

## C    THE PROOF OF LEMMA 1

We first provide the upper bound for the difference of mean vectors $\bar{\mathbf{x}}^{t+1}$ and $\bar{\mathbf{x}}^t$.

**Lemma 8.** *Under the setting of Theorem 1, we have* $\left\| \bar{\mathbf{x}}^{t+1} - \bar{\mathbf{x}}^t \right\|^2 \leq \eta^2 \leq 1/(4L_1^2)$.

*Proof.* The updates in lines 6 and 7 of Algorithm 2 imply

$$
\bar{\mathbf{x}}^{t+1} = \bar{\mathbf{x}}^t - \eta \bar{\mathbf{u}}^t = \bar{\mathbf{x}}^t - \eta \cdot \frac{1}{m} \sum_{i=1}^{m} \frac{\mathbf{v}_i^t}{\|\mathbf{v}_i^t\|}, \tag{8}
$$

where the first step is based on Proposition 2. Therefore, we have

$$
\left\| \bar{\mathbf{x}}^{t+1} - \bar{\mathbf{x}}^t \right\|^2 = \eta^2 \left\| \frac{1}{m} \sum_{i=1}^{m} \frac{\mathbf{v}_i^t}{\|\mathbf{v}_i^t\|} \right\|^2 \leq \frac{\eta^2}{m} \sum_{i=1}^{m} \left\| \frac{\mathbf{v}_i^t}{\|\mathbf{v}_i^t\|} \right\|^2 = \eta^2 \leq \frac{1}{4L_1^2}, \tag{9}
$$

where the first inequality is based on the vector AM-QM inequality and the last step is based on the setting of $\eta = \min \{ \epsilon/(4L_f + 1), 1/(2L_1) \}$.                   □

We then connect the gradient $\nabla f(\bar{\mathbf{x}}^t)$ to $\nabla f(\bar{\mathbf{x}}^{t+1})$ as follows.

**Lemma 9.** *Under the setting of Theorem 1, we have*

$$
\left\| \nabla f(\bar{\mathbf{x}}^{t+1}) \right\| \leq L_f \eta + (1 + L_1 \eta) \left\| \nabla f(\bar{\mathbf{x}}^t) \right\| .
$$

*Proof.* We have

$$
\left\| \nabla f(\bar{\mathbf{x}}^{t+1}) \right\|
$$

$$
\leq \left\| \nabla f(\bar{\mathbf{x}}^{t+1}) - \nabla f(\bar{\mathbf{x}}^t) \right\| + \left\| \nabla f(\bar{\mathbf{x}}^t) \right\|
$$

$$
\leq \left( L_f + L_1 \left\| \nabla f(\bar{\mathbf{x}}^t) \right\| \right) \left\| \bar{\mathbf{x}}^{t+1} - \bar{\mathbf{x}}^t \right\| + \left\| \nabla f(\bar{\mathbf{x}}^t) \right\| \tag{10}
$$

$$
\leq \left( L_f + L_1 \left\| \nabla f(\bar{\mathbf{x}}^t) \right\| \right) \eta + \left\| \nabla f(\bar{\mathbf{x}}^t) \right\|
$$

$$
= L_f \eta + (1 + L_1 \eta) \left\| \nabla f(\bar{\mathbf{x}}^t) \right\| ,
$$

where the first step is based on triangle inequality; the second step is based on Proposition 1; and third last step is based on Lemma 8.                   □

Now we present the proof of Lemma 1.

*Proof.* Applying Lemmas 7, 8 and 9, we have

$$f(\bar{\mathbf{x}}^{t+1}) - f(\bar{\mathbf{x}}^t)$$

$$\overset{(7)}{\leq} \langle \nabla f(\bar{\mathbf{x}}^t), \bar{\mathbf{x}}^{t+1} - \bar{\mathbf{x}}^t \rangle + \frac{L_f + L_1 \|\nabla f(\bar{\mathbf{x}}^t)\|}{2} \|\bar{\mathbf{x}}^{t+1} - \bar{\mathbf{x}}^t\|^2$$

$$\overset{(9)}{\leq} \langle \nabla f(\bar{\mathbf{x}}^t) - \bar{\mathbf{v}}^t, \bar{\mathbf{x}}^{t+1} - \bar{\mathbf{x}}^t \rangle + \langle \bar{\mathbf{v}}^t, \bar{\mathbf{x}}^{t+1} - \bar{\mathbf{x}}^t \rangle + \frac{\eta^2}{2}(L_f + L_1 \|\nabla f(\bar{\mathbf{x}}^t)\|)$$

$$\overset{(8)}{\leq} \|\nabla f(\bar{\mathbf{x}}^t) - \bar{\mathbf{v}}^t\| \|\bar{\mathbf{x}}^{t+1} - \bar{\mathbf{x}}^t\| - \eta \left\langle \bar{\mathbf{v}}^t, \frac{1}{m}\sum_{i=1}^{m} \frac{\mathbf{v}_i^t}{\|\mathbf{v}_i^t\|} \right\rangle + \frac{\eta^2}{2}(L_f + L_1 \|\nabla f(\bar{\mathbf{x}}^t)\|)$$

$$\overset{(9)}{\leq} \eta \|\nabla f(\bar{\mathbf{x}}^t) - \bar{\mathbf{v}}^t\| + \eta \left\langle \bar{\mathbf{v}}^t, \frac{\bar{\mathbf{v}}^t}{\|\bar{\mathbf{v}}^t\|} - \frac{\bar{\mathbf{v}}^t}{\|\bar{\mathbf{v}}^t\|} - \frac{1}{m}\sum_{i=1}^{m} \frac{\mathbf{v}_i^t}{\|\mathbf{v}_i^t\|} \right\rangle + \frac{\eta^2}{2}(L_f + L_1 \|\nabla f(\bar{\mathbf{x}}^t)\|)$$

$$= \eta \|\nabla f(\bar{\mathbf{x}}^t) - \bar{\mathbf{v}}^t\| - \eta \|\bar{\mathbf{v}}^t\| + \eta \left\langle \bar{\mathbf{v}}^t, \frac{\bar{\mathbf{v}}^t}{\|\bar{\mathbf{v}}^t\|} - \frac{1}{m}\sum_{i=1}^{m} \frac{\mathbf{v}_i^t}{\|\mathbf{v}_i^t\|} \right\rangle + \frac{\eta^2}{2}(L_f + L_1 \|\nabla f(\bar{\mathbf{x}}^t)\|) \qquad (11)$$

$$\leq \eta \|\nabla f(\bar{\mathbf{x}}^t) - \bar{\mathbf{v}}^t\| - \eta \|\bar{\mathbf{v}}^t\| + \eta \|\bar{\mathbf{v}}^t\| \left\| \frac{\bar{\mathbf{v}}^t}{\|\bar{\mathbf{v}}^t\|} - \frac{1}{m}\sum_{i=1}^{m} \frac{\mathbf{v}_i^t}{\|\mathbf{v}_i^t\|} \right\| + \frac{\eta^2 L_f}{2} + \frac{\eta^2 L_1}{2} \|\nabla f(\bar{\mathbf{x}}^t)\|$$

$$\leq \eta \|\nabla f(\bar{\mathbf{x}}^t) - \bar{\mathbf{v}}^t\| + \eta(\|\nabla f(\bar{\mathbf{x}}^t) - \bar{\mathbf{v}}^t\| - \|\nabla f(\bar{\mathbf{x}}^t)\|)$$

$$+ \eta \|\bar{\mathbf{v}}^t\| \left\| \frac{\bar{\mathbf{v}}^t}{\|\bar{\mathbf{v}}^t\|} - \frac{1}{m}\sum_{i=1}^{m} \frac{\mathbf{v}_i^t}{\|\mathbf{v}_i^t\|} \right\| + \frac{\eta^2 L_f}{2} + \frac{\eta^2 L_1}{2} \|\nabla f(\bar{\mathbf{x}}^t)\|$$

$$\overset{(6)}{\leq} 2\eta \|\nabla f(\bar{\mathbf{x}}^t) - \bar{\mathbf{v}}^t\| - \eta \left(1 - \frac{\eta L_1}{2}\right) \|\nabla f(\bar{\mathbf{x}}^t)\| + \frac{\eta}{m}\sum_{i=1}^{m} \|\bar{\mathbf{v}}^t - \mathbf{v}_i^t\| + \frac{\eta^2 L_f}{2}$$

$$\leq 2\eta \|\nabla f(\bar{\mathbf{x}}^t) - \bar{\mathbf{v}}^t\| - \eta \left(1 - \frac{\eta L_1}{2}\right) \|\nabla f(\bar{\mathbf{x}}^t)\| + \frac{\eta}{\sqrt{m}} \|\mathbf{V}^t - \mathbf{1}\bar{\mathbf{v}}^t\| + \frac{\eta^2 L_f}{2},$$

where the first inequality is based on Lemma 7 and the fact $\|\bar{\mathbf{x}}^{t+1} - \bar{\mathbf{x}}^t\| \leq 1/(2L_1)$ from Lemma 8; the forth to last line is based on Cauchy–Schwarz inequality; the third to last line is based on the triangle inequality; the last line is based on the vector AM-QM inequality.

We bound the term $\|\nabla f(\bar{\mathbf{x}}^t) - \bar{\mathbf{v}}^t\|$ in equation (11) as

$$\|\nabla f(\bar{\mathbf{x}}^t) - \bar{\mathbf{v}}^t\|^2 = \left\| \frac{1}{m}\sum_{i=1}^{m}(\nabla f_i(\bar{\mathbf{x}}^t) - \mathbf{g}_i^t)) \right\|^2$$

$$\leq 2 \underbrace{\left\| \frac{1}{m}\sum_{i=1}^{m}(\nabla f_i(\bar{\mathbf{x}}^t) - \nabla f_i(\mathbf{x}_i^t)) \right\|^2}_{B_1} + 2 \underbrace{\left\| \frac{1}{m}\sum_{i=1}^{m}(\nabla f_i(\mathbf{x}_i^t) - \mathbf{g}_i^t) \right\|^2}_{B_2}. \qquad (12)$$

We bound term $B_1$ as

$$B_1 = \left\| \frac{1}{m}\sum_{i=1}^{m}(\nabla f_i(\bar{\mathbf{x}}^t) - \nabla f_i(\mathbf{x}_i^t)) \right\|^2$$

$$\leq \frac{1}{m}\sum_{i=1}^{m} \left\| \nabla f_i(\bar{\mathbf{x}}^t) - \nabla f_i(\mathbf{x}_i^t) \right\|^2$$

$$\leq \frac{1}{m}\sum_{i=1}^{m} 2\left(L_0^2 + L_1^2 \|\nabla f_i(\bar{\mathbf{x}}^t)\|^2\right) \|\bar{\mathbf{x}}^t - \mathbf{x}_i^t\|^2 \qquad (13)$$

$$\leq \frac{1}{m}\sum_{i=1}^{m} 2\left(L_0^2 + L_1^2 \left(\|\nabla f(\bar{\mathbf{x}}^t)\| + \zeta\right)^2\right) \|\bar{\mathbf{x}}^t - \mathbf{x}_i^t\|^2$$

$$\leq \frac{1}{m}\left(M_0 + M_1 \|\nabla f(\bar{\mathbf{x}}^t)\|\right)^2 \|\mathbf{X}^t - \mathbf{1}\bar{\mathbf{x}}^t\|^2,$$

where we denote $M_0 = \sqrt{2(L_0^2 + L_1^2\zeta^2)}$ and $M_1 = \sqrt{2}L_1$. Here, the first inequality is based on vector AM-QM inequality; the second inequality is based on Young's inequality, Assumption, 2 and the fact $\|\bar{\mathbf{x}}^t - \mathbf{x}_i^t\| \leq 1/L_1$ from equation (18); the third inequality is based on Assumption 3.

For given $\{\mathbf{x}_i^t\}_{i=1}^m$, we bound expectation of term $B_2$ as

$$
\begin{aligned}
\mathbb{E}[B_2] =& \mathbb{E}\left\|\frac{1}{m}\sum_{i=1}^m (\mathbf{g}_i^t - \nabla f_i(\mathbf{x}_i^t))\right\|^2 \\
=& \frac{1}{m^2}\left(\sum_{i=1}^m \mathbb{E}\left\|\mathbf{g}_i^t - \nabla f_i(\mathbf{x}_i^t)\right\|^2 + \sum_{i=1}^m\sum_{j\neq i}\mathbb{E}\left[\langle \mathbf{g}_i^t - \nabla f_i(\mathbf{x}_i^t), \mathbf{g}_j^t - \nabla f_j(\mathbf{x}_j^t)\rangle\right]\right) \\
=& \frac{1}{m^2}\sum_{i=1}^m \mathbb{E}\left\|\mathbf{g}_i^t - \nabla f_i(\mathbf{x}_i^t)\right\|^2 \leq \frac{1}{m^2}\sum_{i=1}^m \frac{\sigma^2}{b} = \frac{\sigma^2}{mb},
\end{aligned}
\tag{14}
$$

where the second step is based on the fact that $\mathbf{g}_i^t$ and $\mathbf{g}_j^t$ are independent for $i \neq j$ which implies their covariance $\mathbb{E}[\langle \mathbf{g}_i^t - \nabla f_i(\mathbf{x}_i^t), \mathbf{g}_j^t - \nabla f_i(\mathbf{x}_j^t)\rangle]$ is zero; the inequality is based on the setting $\mathbf{g}_i^t = \frac{1}{b}\sum_{k=1}^b \nabla F_i(\mathbf{x}_i^t; \boldsymbol{\xi}_{i,k}^t)$ in line 9 of Algorithm 2 and Assumption 4.

Combining results (12), (13) and (14), we have

$$
\begin{aligned}
\mathbb{E}\left\|\nabla f(\bar{\mathbf{x}}^t) - \bar{\mathbf{v}}^t\right\| \leq& \sqrt{\frac{2}{m}\left(M_0 + M_1\left\|\nabla f(\bar{\mathbf{x}}^t)\right\|\right)^2 \left\|\mathbf{X}^t - \mathbf{1}\bar{\mathbf{x}}^t\right\|^2 + \frac{\sigma^2}{mb}} \\
\leq& \sqrt{\frac{2}{m}}\left(M_0 + M_1\left\|\nabla f(\bar{\mathbf{x}}^t)\right\|\right)\left\|\mathbf{X}^t - \mathbf{1}\bar{\mathbf{x}}^t\right\| + \frac{\sigma}{\sqrt{mb}},
\end{aligned}
\tag{15}
$$

for given $\mathbf{X}^t$, where the first step is based on the Jensen's inequality and the second step is based on the elementary inequality $\sqrt{a^2 + b^2} \leq a + b$ for $a, b > 0$.

Combining equations (11) and (15), we finish the proof. $\qquad\square$

## D   THE PROOF OF LEMMA 2

*Proof.* Recall that line 6 of Algorithm 2 set

$$
\mathbf{U}^t = \left[\frac{\mathbf{v}_1^t}{\|\mathbf{v}_1^t\|} \quad \cdots \quad \frac{\mathbf{v}_1^t}{\|\mathbf{v}_m^t\|}\right]^\top,
$$

which implies $\|\mathbf{U}^t\|^2 = m$. Based on equation (8), we have

$$
\begin{aligned}
&\left\|\mathbf{X}^{t+1} - \mathbf{1}\bar{\mathbf{x}}^{t+1}\right\| \\
=& \left\|\text{AccGossip}(\mathbf{X}^t - \eta\mathbf{U}^t, K) - \frac{1}{m}\mathbf{1}\mathbf{1}^\top\text{AccGossip}(\mathbf{X}^t - \eta\mathbf{U}^t, K)\right\| \\
\leq& \rho\left\|\mathbf{X}^t - \eta\mathbf{U}^t - \frac{1}{m}\mathbf{1}\mathbf{1}^\top(\mathbf{X}^t - \eta\mathbf{U}^t)\right\| \\
=& \rho\left\|\mathbf{X}^t - \eta\mathbf{U}^t - \mathbf{1}(\bar{\mathbf{x}}^t - \eta\bar{\mathbf{u}}^t)\right\| \\
\leq& \rho\left(\left\|\mathbf{X}^t - \mathbf{1}\bar{\mathbf{x}}^t\right\| + \eta\left\|\mathbf{U}^t - \mathbf{1}\bar{\mathbf{u}}^t\right\|\right) \\
\leq& \rho\left(\left\|\mathbf{X}^t - \mathbf{1}\bar{\mathbf{x}}^t\right\| + m\eta\right),
\end{aligned}
\tag{16}
$$

where the first inequality is based on Proposition 2; the second inequality is based on the triangle inequality; the last step applies Lemma 4 which leads to

$$
\left\|\mathbf{U}^t - \mathbf{1}\bar{\mathbf{u}}^t\right\|^2 \leq \left\|\mathbf{U}^t\right\|^2 = m.
$$

Based on equation (16), we have

$$
\left\|\mathbf{X}^t - \mathbf{1}\bar{\mathbf{x}}^t\right\| \leq \rho^t\left(\left\|\mathbf{X}^0 - \mathbf{1}\bar{\mathbf{x}}^0\right\| - \frac{\rho m\eta}{1-\rho}\right) + \frac{\rho m\eta}{1-\rho} \leq \frac{\rho m\eta}{1-\rho},
$$

where the second inequality is based on the initialization such that $\mathbf{x}_i^0 = \bar{\mathbf{x}}^0$ for each $i$.

$\qquad\square$

# E  THE PROOF OF LEMMA 3

*Proof.* We denote

$$\nabla \mathbf{f}(\mathbf{X}^t) = \begin{bmatrix} \nabla f_1(\mathbf{x}_1^t)^\top \\ \vdots \\ \nabla f_m(\mathbf{x}_m^t)^\top \end{bmatrix} \in \mathbb{R}^{m \times d} \quad \text{and} \quad \nabla \mathbf{f}(\mathbf{X}^{t+1}) = \begin{bmatrix} \nabla f_1(\mathbf{x}_1^{t+1})^\top \\ \vdots \\ \nabla f_m(\mathbf{x}_m^{t+1})^\top \end{bmatrix} \in \mathbb{R}^{m \times d}.$$

The update in line 10 of Algorithm 2 implies

$$
\begin{aligned}
&\left\| \mathbf{V}^{t+1} - \mathbf{1}\bar{\mathbf{v}}^{t+1} \right\| \\
&= \left\| \text{AccGossip}(\mathbf{V}^t + \mathbf{G}^{t+1} - \mathbf{G}^t), K) - \frac{1}{m}\mathbf{1}\mathbf{1}^\top \text{AccGossip}(\mathbf{V}^t + \mathbf{G}^{t+1} - \mathbf{G}^t), K) \right\| \\
&\le \rho \left\| \mathbf{V}^t + \mathbf{G}^{t+1} - \mathbf{G}^t - \frac{1}{m}\mathbf{1}\mathbf{1}^\top (\mathbf{V}^t + \mathbf{G}^{t+1} - \mathbf{G}^t) \right\| \\
&\le \rho \left\| \mathbf{V}^t - \mathbf{1}\bar{\mathbf{v}}^t \right\| + \rho \left\| \mathbf{G}^{t+1} - \mathbf{G}^t - \frac{1}{m}\mathbf{1}\mathbf{1}^\top (\mathbf{G}^{t+1} - \mathbf{G}^t) \right\| \\
&\le \rho \left\| \mathbf{V}^t - \mathbf{1}\bar{\mathbf{v}}^t \right\| + \rho \left\| \mathbf{G}^{t+1} - \mathbf{G}^t \right\| \\
&\le \rho \left\| \mathbf{V}^t - \mathbf{1}\bar{\mathbf{v}}^t \right\| + \rho \underbrace{\left\| \nabla \mathbf{f}(\mathbf{X}^{t+1}) - \nabla \mathbf{f}(\mathbf{X}^t) \right\|}_{C_1} + \rho \underbrace{\left\| \mathbf{G}^{t+1} - \nabla \mathbf{f}(\mathbf{X}^{t+1}) \right\|}_{C_2} + \rho \underbrace{\left\| \mathbf{G}^t - \nabla \mathbf{f}(\mathbf{X}^t) \right\|}_{C_3},
\end{aligned}
\tag{17}
$$

where the first three step follows the derivation of equation (16); the fourth step is based on Lemma 4; the last step is based on triangle inequality.

We first consider term $C_1$. The setting of $K$ in this lemma implies

$$\rho \le \frac{1}{1 + m\eta L_1},$$

which leads to

$$\left\| \mathbf{x}_t^i - \bar{\mathbf{x}}^t \right\| \le \left\| \mathbf{X}^t - \mathbf{1}\bar{\mathbf{x}}^t \right\| \le \frac{\rho m \eta}{1 - \rho} \le \frac{1}{L_1}. \tag{18}$$

Therefore, we have

$$
\begin{aligned}
&\left\| \nabla f(\mathbf{x}_i^{t+1}) - \nabla f(\mathbf{x}_i^t) \right\| \\
&\le \left\| \nabla f(\mathbf{x}_i^{t+1}) - \nabla f(\bar{\mathbf{x}}^{t+1}) \right\| + \left\| \nabla f(\bar{\mathbf{x}}^{t+1}) - \nabla f(\bar{\mathbf{x}}^t) \right\| + \left\| \nabla f(\bar{\mathbf{x}}^t) - \nabla f(\mathbf{x}_i^t) \right\| \\
&\le \left( L_f + L_1 \left\| \nabla f(\bar{\mathbf{x}}^{t+1}) \right\| \right) \left\| \mathbf{x}_i^{t+1} - \bar{\mathbf{x}}^{t+1} \right\| + \left( L_f + L_1 \left\| \nabla f(\bar{\mathbf{x}}^t) \right\| \right) \left\| \bar{\mathbf{x}}^{t+1} - \bar{\mathbf{x}}^t \right\| \\
&\quad + \left( L_f + L_1 \left\| \nabla f(\bar{\mathbf{x}}^t) \right\| \right) \left\| \mathbf{x}_i^t - \bar{\mathbf{x}}^t \right\|,
\end{aligned}
$$

where the first step is based on the triangle inequality and the last step is based on Proposition 1 and equation (18). Summing over above result with $i = 1, \ldots, m$, we have

$$
\begin{aligned}
C_1 &= \sum_{i=1}^m \left\| \nabla f(\mathbf{x}_i^{t+1}) - \nabla f(\mathbf{x}_i^t) \right\|^2 \\
&\le 3 \left( L_f + L_1 \left\| \nabla f(\bar{\mathbf{x}}^{t+1}) \right\| \right)^2 \sum_{i=1}^m \left\| \mathbf{x}_i^{t+1} - \bar{\mathbf{x}}^{t+1} \right\|^2 \\
&\quad + 3 \left( L_f + L_1 \left\| \nabla f(\bar{\mathbf{x}}^t) \right\| \right)^2 \sum_{i=1}^m \left\| \bar{\mathbf{x}}^{t+1} - \bar{\mathbf{x}}^t \right\|^2 + 3 \left( L_f + L_1 \left\| \nabla f(\bar{\mathbf{x}}^t) \right\| \right)^2 \sum_{i=1}^m \left\| \mathbf{x}_i^t - \bar{\mathbf{x}}^t \right\|^2 \\
&\le 3 \left( L_f + L_1 \left\| \nabla f(\bar{\mathbf{x}}^{t+1}) \right\| \right)^2 \left\| \mathbf{X}^{t+1} - \mathbf{1}\bar{\mathbf{x}}^{t+1} \right\|^2 \\
&\quad + 3m \left( L_f + L_1 \left\| \nabla f(\bar{\mathbf{x}}^t) \right\| \right)^2 \eta^2 + 3 \left( L_f + L_1 \left\| \nabla f(\bar{\mathbf{x}}^t) \right\| \right)^2 \left\| \mathbf{X}^t - \mathbf{1}\bar{\mathbf{x}}^t \right\|^2,
\end{aligned}
\tag{19}
$$

where the last step is based on Lemma 8.

Substituting equations (16) and (10) into equation (19), we have

$$
\begin{aligned}
&\left\| \nabla \mathbf{f}(\mathbf{X}^{t+1}) - \nabla \mathbf{f}(\mathbf{X}^t) \right\|^2 \\
&\le 3\rho^2 \left( L_f + L_1 \left( L_f \eta + (L_1 \eta + 1) \left\| \nabla f(\bar{\mathbf{x}}^t) \right\| \right) \right)^2 \left( \left\| \mathbf{X}^t - \mathbf{1}\bar{\mathbf{x}}^t \right\| + m\eta \right)^2 \\
&\quad + 3 \left( L_f + L_1 \left\| \nabla f(\bar{\mathbf{x}}^t) \right\| \right)^2 \left\| \mathbf{X}^t - \mathbf{1}\bar{\mathbf{x}}^t \right\|^2 + 3m \left( L_f + L_1 \left\| \nabla f(\bar{\mathbf{x}}^t) \right\| \right)^2 \eta^2 \\
&\le \left( M_2 + M_3 \left\| \nabla f(\bar{\mathbf{x}}^t) \right\| \right)^2 \left( \left\| \mathbf{X}^t - \mathbf{1}\bar{\mathbf{x}}^t \right\| + m\eta \right)^2,
\end{aligned}
\tag{20}
$$

where we take $M_2 = 3(\rho + 1)L_f + 3\rho L_f L_1 \eta$ and $M_3 = 3\rho L_1(L_1\eta + 1) + 3L_1$.

For the term $C_2$, we have

$$
\begin{aligned}
&\mathbb{E}\left[\left\|\mathbf{G}^{t+1} - \nabla\mathbf{f}(\mathbf{X}^{t+1})\right\|^2\right] \\
=&\mathbb{E}\left[\sum_{i=1}^m \left\|\mathbf{g}_i^{t+1} - \nabla f_i(\mathbf{x}_i^{t+1})\right\|^2\right] \\
=&\mathbb{E}\left[\sum_{i=1}^m \left\|\frac{1}{b}\sum_{k=1}^b \nabla F_i(\mathbf{x}_i^{t+1}; \boldsymbol{\xi}_{i,k}^{t+1}) - \nabla f_i(\mathbf{x}_i^{t+1})\right\|^2\right] \\
\leq& \frac{m\sigma^2}{b},
\end{aligned}
\tag{21}
$$

for given $\{\mathbf{x}_i^{t+1}\}_{i=1}^m$, where the inequality is based on Assumption 4. Consequently, we have

$$
\begin{aligned}
C_2 =&\mathbb{E}\left[\left\|\mathbf{G}^{t+1} - \nabla\mathbf{f}(\mathbf{X}^{t+1})\right\|\right] \\
=&\sqrt{(\mathbb{E}\left\|\mathbf{G}^{t+1} - \nabla\mathbf{f}(\mathbf{X}^{t+1})\right\|)^2} \\
\leq&\sqrt{\mathbb{E}\left[\left\|\mathbf{G}^{t+1} - \nabla\mathbf{f}(\mathbf{X}^{t+1})\right\|^2\right]} \\
\overset{(21)}{\leq}& \frac{\sqrt{m}\sigma}{\sqrt{b}},
\end{aligned}
\tag{22}
$$

where the first inequality is based on Jensen's inequality.

Similarly, the term $C_3$ holds

$$
C_3 = \mathbb{E}\left[\left\|\mathbf{G}^t - \nabla\mathbf{f}(\mathbf{X}^t)\right\|\right] \leq \frac{\sqrt{m}\sigma}{\sqrt{b}}.
\tag{23}
$$

Combining the results of (17), (20), (22), and (23), we have

$$
\begin{aligned}
&\mathbb{E}\left[\left\|\mathbf{V}^{t+1} - \mathbf{1}\bar{\mathbf{v}}^{t+1}\right\|\right] \\
\leq&\rho\left\|\mathbf{V}^t - \mathbf{1}\bar{\mathbf{v}}^t\right\| + \rho\left(M_2 + M_3\left\|\nabla f(\bar{\mathbf{x}}^t)\right\|\right)\left(\left\|\mathbf{X}^t - \mathbf{1}\bar{\mathbf{x}}^t\right\| + m\eta\right) + \frac{2\rho\sqrt{m}\sigma}{\sqrt{b}} \\
\leq&\rho\left(\left\|\mathbf{V}^t - \mathbf{1}\bar{\mathbf{v}}^t\right\| + \left(M_2 + M_3\left\|\nabla f(\bar{\mathbf{x}}^t)\right\|\right)\left\|\mathbf{X}^t - \mathbf{1}\bar{\mathbf{x}}^t\right\| + \left(M_2 + M_3\left\|\nabla f(\bar{\mathbf{x}}^t)\right\|\right)m\eta + \frac{2\sqrt{m}\sigma}{\sqrt{b}}\right),
\end{aligned}
$$

for given $\{\mathbf{x}_i^{t+1}\}_{i=1}^m$ and $\{\mathbf{v}_i^{t+1}\}_{i=1}^m$, which finishes the proof. $\qquad\square$

## F  THE PROOF OF THEOREM 1

For given $\mathbf{X}^t$ and $\mathbf{V}^t$, we apply Lemmas 1, 2, 3, and 9 to achieve

$$
\begin{aligned}
&\mathbb{E}[\Phi^{t+1}] \\
=&\mathbb{E}\left[f(\bar{\mathbf{x}}^{t+1}) + \frac{3\eta}{\sqrt{m}}\left(M_0 + M_1\left\|\nabla f(\bar{\mathbf{x}}^{t+1})\right\|\right)\left\|\mathbf{X}^{t+1} - \mathbf{1}\bar{\mathbf{x}}^{t+1}\right\| + \frac{2\eta}{\sqrt{m}}\left\|\mathbf{V}^{t+1} - \mathbf{1}\bar{\mathbf{v}}^{t+1}\right\|\right] \\
\leq&f(\bar{\mathbf{x}}^t) - \eta\left(1 - \frac{\eta L_1}{2}\right)\left\|\nabla f(\bar{\mathbf{x}}^t)\right\| \\
&+ \frac{2\sqrt{2}\eta}{\sqrt{m}}\left(M_0 + M_1\left\|\nabla f(\bar{\mathbf{x}}^t)\right\|\right)\left\|\mathbf{X}^t - \mathbf{1}\bar{\mathbf{x}}^t\right\| + \frac{\eta}{\sqrt{m}}\left\|\mathbf{V}^t - \mathbf{1}\bar{\mathbf{v}}^t\right\| + \frac{2\eta\sigma}{\sqrt{mb}} + \frac{\eta^2 L_f}{2} \\
&+ \frac{3\eta\rho}{\sqrt{m}}\left(M_0 + M_1\left(L_f\eta + (1 + L_1\eta)\left\|\nabla f(\bar{\mathbf{x}}^t)\right\|\right)\right)\left(\left\|\mathbf{X}^t - \mathbf{1}\bar{\mathbf{x}}^t\right\| + m\eta\right) \\
&+ \frac{2\eta\rho}{\sqrt{m}}\left(\left\|\mathbf{V}^t - \mathbf{1}\bar{\mathbf{v}}^t\right\| + \left(M_2 + M_3\left\|\nabla f(\bar{\mathbf{x}}^t)\right\|\right)\left\|\mathbf{X}^t - \mathbf{1}\bar{\mathbf{x}}^t\right\| + \left(M_2 + M_3\left\|\nabla f(\bar{\mathbf{x}}^t)\right\|\right)m\eta + \frac{2\sqrt{m}\sigma}{\sqrt{b}}\right)
\end{aligned}
$$

$$=f(\bar{\mathbf{x}}^t) + \left(\frac{2\sqrt{2}M_0\eta}{\sqrt{m}} + \rho\Big(\frac{3\eta(M_0 + M_1 L_f\eta)}{\sqrt{m}} + \frac{2\eta M_2}{\sqrt{m}}\Big)\right)\left\|\mathbf{X}^t - \mathbf{1}\bar{\mathbf{x}}^t\right\|$$

$$+ \left(\frac{2\sqrt{2}M_1\eta}{\sqrt{m}} + \rho\left(\frac{3\eta M_1(1 + L_1\eta)}{\sqrt{m}} + \frac{2\eta M_3}{\sqrt{m}}\right)\right)\left\|\nabla f(\bar{\mathbf{x}}^t)\right\|\left\|\mathbf{X}^t - \mathbf{1}\bar{\mathbf{x}}^t\right\| + \left(\frac{\eta}{\sqrt{m}} + \frac{2\rho\eta}{\sqrt{m}}\right)\left\|\mathbf{V}^t - \mathbf{1}\bar{\mathbf{v}}^t\right\|$$

$$- \eta\left(1 - \frac{\eta L_1}{2} - \rho\Big(3\sqrt{m}\eta M_1(1 + L_1\eta) + 2\sqrt{m}\eta M_3\Big)\right)\left\|\nabla f(\bar{\mathbf{x}}^t)\right\|$$

$$+ \frac{2\eta\sigma}{\sqrt{mb}} + \frac{\eta^2 L_f}{2} + \rho\Big(3\sqrt{m}\eta^2(M_0 + M_1 L_f\eta) + 2\sqrt{m}\eta^2 M_2 + \frac{4\eta\sigma}{\sqrt{b}}\Big)$$

$$\leq f(\bar{\mathbf{x}}^t) + \frac{3\eta}{\sqrt{m}}\left(M_0 + M_1\left\|\nabla f(\bar{\mathbf{x}}^t)\right\|\right)\left\|\mathbf{X}^t - \mathbf{1}\bar{\mathbf{x}}^t\right\| + \frac{2\eta}{\sqrt{m}}\left\|\mathbf{V}^t - \mathbf{1}\bar{\mathbf{v}}^t\right\|$$

$$- \eta\left(\frac{7}{8} - \frac{\eta L_1}{2}\right)\left\|\nabla f(\bar{\mathbf{x}}^t)\right\| + \frac{3\eta^2 L_f}{4}$$

$$= \Phi^t - \eta\left(\frac{7}{8} - \frac{\eta L_1}{2}\right)\left\|\nabla f(\bar{\mathbf{x}}^t)\right\| + \frac{3\eta^2 L_f}{4}$$

$$\leq \Phi^t - \frac{5\eta}{8}\left\|\nabla f(\bar{\mathbf{x}}^t)\right\| + \frac{3\eta^2 L_f}{4},$$

where the second inequality requires the condition

$$\begin{cases} 2\sqrt{2}M_0 + \rho(3(M_0 + M_1 L_f\eta) + 2M_2) \leq 3M_0, \\ 2\sqrt{2}M_1 + \rho\left(3M_1(1 + L_1\eta) + 2M_3\right) \leq 3M_1, \\ \eta + 2\rho\eta \leq 2\eta, \\ \rho\Big(3\sqrt{m}\eta M_1(1 + L_1\eta) + 2\sqrt{m}\eta M_3\Big) \leq \frac{1}{8}, \\ \frac{2\sigma}{\sqrt{mb}} + \rho\Big(3\sqrt{m}\eta(M_0 + M_1 L_f\eta) + 2\sqrt{m}\eta M_2 + \frac{4\eta\sigma}{\sqrt{b}}\Big) \leq \frac{\eta L_f}{4} \end{cases}$$

that holds by taking $b \geq \left\lceil 256\sigma^2/(m\eta^2 L_f^2) \right\rceil$ and

$$\rho \leq \min\left\{ \frac{1}{1 + m\eta L_1}, \frac{(3 - 2\sqrt{2})M_0}{3(M_0 + M_1 L_f\eta) + 2M_2}, \frac{(3 - 2\sqrt{2})M_1}{3M_1(1 + L_1\eta) + 2M_3}, \frac{1}{2}, \right.$$

$$\left. \frac{1}{8(3\sqrt{m}\eta M_1(1 + L_1\eta) + 2\sqrt{m}\eta M_3)}, \frac{\eta L_f}{8(3\sqrt{m}\eta(M_0 + M_1 L_f\eta) + 2\sqrt{m}\eta M_2 + 4\eta\sigma/\sqrt{b})} \right\}.$$

Recall that we have $L_1\eta \leq 1/2$, $M_0 = \Theta(L_f)$, $M_1 = \Theta(L_1)$, $M_2 \leq \Theta(L_f)$, and $M_3 \leq \Theta(L_1)$, which means we require

$$\rho \leq \mathcal{O}\left(\frac{1}{\sqrt{m}(1 + \eta)}\right).$$

This can be achieved by taking

$$K = \mathcal{O}\left(\frac{\log(\sqrt{m}(1 + \eta))}{\sqrt{\gamma}}\right).$$

Following the setting $\eta = \min\{\epsilon/(4L_f + 1), 1/(2L_1)\}$, we have

$$\mathbb{E}\left[\left\|\nabla f(\bar{\mathbf{x}}^t)\right\|\right] \leq \frac{\mathbb{E}[8(\Phi^t - \Phi^{t+1})]}{5\eta} + \frac{6\eta L_f}{5}.$$

Taking the average on above inequalities with $t = 0, \ldots, T - 1$, we have

$$\mathbb{E}\left[\frac{1}{T}\sum_{t=0}^{T-1}\left\|\nabla f(\bar{\mathbf{x}}^t)\right\|\right] \leq \mathbb{E}\left[\frac{8(\Phi^0 - \Phi^T)}{5\eta T} + \frac{6\eta L_f}{5}\right] \leq \frac{8\Delta_\Phi}{5\eta T} + \frac{6\eta L_f}{5},$$

where the last step holds due to

$$\mathbb{E}[\Phi^0 - \Phi^T]$$

$$=\mathbb{E}\left[f(\bar{\mathbf{x}}^0) + \frac{3\eta}{\sqrt{m}}\left(M_0 + M_1 \left\|\nabla f(\bar{\mathbf{x}}^0)\right\|\right)\left\|\mathbf{X}^0 - \mathbf{1}\bar{\mathbf{x}}^0\right\| + \frac{2\eta}{\sqrt{m}}\left\|\mathbf{V}^0 - \mathbf{1}\bar{\mathbf{v}}^0\right\|\right.$$

$$\left. - \left(f(\bar{\mathbf{x}}^T) + \frac{3\eta}{\sqrt{m}}\left(M_0 + M_1 \left\|\nabla f(\bar{\mathbf{x}}^T)\right\|\right)\left\|\mathbf{X}^T - \mathbf{1}\bar{\mathbf{x}}^T\right\| + \frac{2\eta}{\sqrt{m}}\left\|\mathbf{V}^T - \mathbf{1}\bar{\mathbf{v}}^T\right\|\right)\right]$$

$$\leq f(\bar{\mathbf{x}}^0) - \inf_{\mathbf{x}\in\mathbb{R}^d} f(\mathbf{x}) + \frac{2\eta}{\sqrt{m}}\mathbb{E}\left[\left\|\mathbf{V}^0 - \mathbf{1}\bar{\mathbf{v}}^0\right\|\right] = \Delta_\Phi.$$

We consider the following case for the choice of $\eta$:

1. In the case of $\epsilon/(4L_f + 1) \leq 1/(2L_1)$, we follow the setting

$$\eta = \frac{\epsilon}{4L_f + 1}, \quad b \geq \left\lceil\frac{256(4L_f + 1)^2\sigma^2}{mL_f^2\epsilon^2}\right\rceil, \quad \text{and} \quad T \geq \frac{8\Delta_\Phi}{\eta\epsilon} = \frac{8(4L_f + 1)\Delta_\Phi}{\epsilon^2} \quad (24)$$

to obtain

$$\mathbb{E}\left[\frac{1}{T}\sum_{t=0}^{T-1}\left\|\nabla f(\bar{\mathbf{x}}^t)\right\|\right] \leq \frac{\epsilon}{5} + \frac{3\epsilon}{10} = \frac{\epsilon}{2}.$$

2. In the case of $1/(2L_1) \leq \epsilon/(4L_f + 1)$, we follow the setting

$$\eta = \frac{1}{2L_1}, \quad b \geq \left\lceil\frac{1024L_1^2\sigma^2}{mL_f^2}\right\rceil, \quad \text{and} \quad T \geq \frac{16L_1\Delta_\Phi}{\epsilon}. \quad (25)$$

to obtain

$$\mathbb{E}\left[\frac{1}{T}\sum_{t=0}^{T-1}\left\|\nabla f(\bar{\mathbf{x}}^t)\right\|\right] \leq \frac{\epsilon}{5} + \frac{3\epsilon}{10} = \frac{\epsilon}{2}.$$

Hence, we finish the proof.

## G  THE PROOF OF COROLLARY 1

*Proof.* We first provide the upper bound for $\mathbb{E}\left[\left\|\mathbf{V}^0 - \mathbf{1}\bar{\mathbf{v}}^0\right\|\right]$. Let $\bar{\mathbf{g}}^0 = \frac{1}{m}\mathbf{1}^\top\mathbf{G}^0$, then we have

$$\mathbb{E}\left[\left\|\mathbf{V}^0 - \mathbf{1}\bar{\mathbf{v}}^0\right\|^2\right] \leq \mathbb{E}\left[\hat{\rho}^2\left\|\mathbf{G}^0 - \mathbf{1}\bar{\mathbf{g}}^0\right\|^2\right]$$

$$\leq \mathbb{E}\left[\hat{\rho}^2\left\|\mathbf{G}^0\right\|^2\right] = \hat{\rho}^2\sum_{i=1}^m\mathbb{E}\left\|\frac{1}{b}\sum_{k=1}^b\nabla F_i(\mathbf{x}_i^0; \boldsymbol{\xi}_{i,k}^0)\right\|^2$$

$$=2\hat{\rho}^2\sum_{i=1}^m\left(\mathbb{E}\left[\left\|\frac{1}{b}\sum_{k=1}^b\nabla F_i(\mathbf{x}_i^0; \boldsymbol{\xi}_{i,k}^0) - \nabla f_i(\mathbf{x}_i^0)\right\|^2 + \left\|\nabla f_i(\mathbf{x}_i^0)\right\|^2\right]\right)$$

$$=2\hat{\rho}^2\sum_{i=1}^m\left(\frac{\sigma^2}{b} + \left\|\nabla f_i(\mathbf{x}_i^0)\right\|^2\right)$$

$$=2\hat{\rho}^2\left(\frac{m\sigma^2}{b} + \sum_{i=1}^m\left\|\nabla f_i(\mathbf{x}_i^0)\right\|^2\right).$$

Hence, we require

$$\hat{\rho} \leq \frac{\sqrt{m}\Delta_f}{2\sqrt{2}\eta\sqrt{m\sigma^2/b + \sum_{i=1}^m\left\|\nabla f_i(\mathbf{x}_i^0)\right\|^2}}$$

to guarantee

$$\mathbb{E}\left[\left\|\mathbf{V}^0 - \mathbf{1}\bar{\mathbf{v}}^0\right\|\right] \leq \sqrt{\mathbb{E}\left[\left\|\mathbf{V}^0 - \mathbf{1}\bar{\mathbf{v}}^0\right\|^2\right]} \leq \frac{\sqrt{m}\Delta_f}{2\eta}, \tag{26}$$

where the first step is based on Jensen's inequality. Consequently, we have

$$\Delta_\Phi = f(\bar{\mathbf{x}}^0) - \inf_{\mathbf{x}\in\mathbb{R}^d} f(\mathbf{x}) + \frac{2\eta}{\sqrt{m}}\mathbb{E}\left[\left\|\mathbf{V}^0 - \mathbf{1}\bar{\mathbf{v}}^0\right\|\right] \leq 2\Delta_f$$

and the requirement of above $\hat{\rho}$ can be satasfied by taking

$$\hat{K} \geq \left\lceil \frac{1}{\sqrt{\gamma}} \log \frac{2\sqrt{2}\eta\sqrt{m\sigma^2/b + \sum_{i=1}^m \left\|\nabla f_i(\mathbf{x}_i^0)\right\|^2}}{\sqrt{m}\Delta_f} \right\rceil = \tilde{\mathcal{O}}\left(\frac{1}{\sqrt{\gamma}}\right).$$

Therefore, the term $\Delta_\Phi$ in the setting of $T$ presented in Theorem 1 can be replaced by $2\Delta_f$.

We then consider how to making the gradient small on each node. Specifically, we have

$$\begin{aligned}
\mathbb{E}\left[\left\|\nabla f(\hat{\mathbf{x}}_i)\right\|\right] =&\, \mathbb{E}\left[\frac{1}{T}\sum_{t=0}^{T-1}\left\|\nabla f(\mathbf{x}_i^t)\right\|\right] \\
\leq&\, \mathbb{E}\left[\frac{1}{T}\sum_{t=0}^{T-1}\left\|\nabla f(\mathbf{x}_i^t) - \nabla f(\bar{\mathbf{x}}^t)\right\| + \frac{1}{T}\sum_{t=0}^{T-1}\left\|\nabla f(\bar{\mathbf{x}}^t)\right\|\right] \\
\leq&\, \mathbb{E}\left[\frac{1}{T}\sum_{t=0}^{T-1}\left(L_f + L_1\left\|\nabla f(\bar{\mathbf{x}}^t)\right\|\right)\left\|\mathbf{x}_i^t - \bar{\mathbf{x}}^t\right\| + \frac{1}{T}\sum_{t=0}^{T-1}\left\|\nabla f(\bar{\mathbf{x}}^t)\right\|\right] \\
\leq&\, \mathbb{E}\left[\frac{1}{T}\sum_{t=0}^{T-1}\left(L_f + L_1\left\|\nabla f(\bar{\mathbf{x}}^t)\right\|\right)\left\|\mathbf{X}^t - \mathbf{1}\bar{\mathbf{x}}^t\right\|\right] + \frac{\epsilon}{2} \\
\leq&\, \mathbb{E}\left[\frac{1}{T}\sum_{t=0}^{T-1}\frac{\rho m\eta\left(L_f + L_1\left\|\nabla f(\bar{\mathbf{x}}^t)\right\|\right)}{1-\rho}\right] + \frac{\epsilon}{2} \\
=&\, \frac{\rho m\eta\left(L_f + \mathbb{E}\left[\frac{L_1}{T}\sum_{t=0}^{T-1}\left\|\nabla f(\bar{\mathbf{x}}^t)\right\|\right]\right)}{1-\rho} + \frac{\epsilon}{2} \\
\leq&\, \frac{\rho m\eta\left(L_f + L_1\epsilon/2\right)}{1-\rho} + \frac{\epsilon}{2} \leq \epsilon,
\end{aligned}$$

where the first step is based on the triangle inequality; the second step is based on Proposition 1 and equation (18); the third step is based on Theorem 1; the fourth step is based on Lemma 2; the last line requires

$$\rho \leq \frac{\epsilon}{2m\eta L_f + (m\eta L_1 + 1)\epsilon},$$

which can be satisfies by taking $K = \tilde{\mathcal{O}}(\sqrt{1/\gamma})$.

In summary, the communication complexity is

$$\hat{K} + TK = \tilde{\mathcal{O}}\left(\left(\frac{L_f}{\epsilon^2} + \frac{L_1}{\epsilon}\right)\frac{\Delta_f}{\sqrt{\gamma}}\right).$$

Following equations (24), (25), and (26), the sample complexity on each node is

$$Tb = \mathcal{O}\left(\frac{1}{m}\left(\frac{L_f\sigma^2\Delta}{\epsilon^4} + \frac{\sigma^2}{\epsilon^2} + \frac{L_1^3\sigma^2\Delta}{L_f^2\epsilon} + \frac{L_1^2\sigma^2}{L_f^2}\right)\right),$$

where the terms $\sigma^2/\epsilon^2$ and $L_1^2\sigma^2/L_f^2$ come from the initialization on $\mathbf{G}^0$ in line 3 of Algorithm 2. $\quad\square$

## H  THE PROOF OF PROPOSITION 3

*Proof.* We can verify that the (second-order) derivatives of functions $f_1$, $f_2$, and $f$ have the form of

$$f_1'(x) = L\exp(Lx), \quad f_2'(x) = -L\exp(-Lx), \quad \text{and} \quad f'(x) = \frac{L\exp(Lx) - L\exp(-Lx)}{2}.$$

Note that the function

$$h(z) = \frac{|1 - \exp(Lz)|}{L|z|}$$

is increasing. Taking $z = y - x \neq 0$, we have

$$\frac{|1 - \exp(L(y-x))|}{L|y-x|} = h(y-x) \leq h\left(\frac{1}{L_1}\right) = \frac{|1 - \exp(L/L_1)|}{L/L_1} \tag{27}$$

for all $x, y \in \mathbb{R}$ with $|y - x| \in [-1/L_1, 1/L_1]$.

For the function $f_1$, we have

$$\begin{aligned}
&|f_1'(x) - f_1'(y)|\\
=&|L\exp(Lx) - L\exp(Ly)|\\
=&L\exp(Lx)|1 - \exp(L(y-x))|\\
\leq&L\exp(Lx)\cdot L_1|1 - \exp(L/L_1)||y-x|\\
=&L\exp(Lx)|y-x|L_1\\
=&L_1 f_1'(x)|x-y|,
\end{aligned}$$

where the inequality is based on equation (27) and the last two steps are based on $f_1'(x) = L\exp(Lx)$ and the setting $L_1 = L/\log 2$. Therefore, we have shown that $f_1$ is $(0, L_1)$-smooth. Similarly, we can also show that $f_2$ is $(0, L_1)$-smooth.

Let $x = 0$ and $y = (\log 2)/L$, which holds $|x - y| = 1/L_1$. We also have

$$\begin{aligned}
&|f'(x) - f'(y)|\\
=&\frac{L}{2}\left|\exp(Lx) - \exp(-Lx) - (\exp(Ly) - \exp(-Ly))\right|\\
=&\frac{L}{2}\left|\exp(0) - \exp(0) - \left(2 - \frac{1}{2}\right)\right| = \frac{3L}{4}
\end{aligned}$$

and

$$L_1 f'(x)|y - x| = \frac{L_1 L(\exp(Lx) - \exp(-Lx))|y-x|}{2} = 0.$$

Therefore, the inequality $|f'(x) - f'(y)| \leq L_1 f'(x)|y-x|$ does not hold, which means the function $f$ is not $(0, L_1)$-smooth. $\qquad\square$

## I  ADDITIONAL EXPERIMENTS

In this section, we provide numerical results for fine-tuning large language models (LLMs). Specifically, we fine-tune nanoGPT (124M parameter)[3] on the Shakespeare dataset. Similar to the setting in Section 6, we perform experiments on two network topologies with 4 agents, i.e., the Erdös-Rényi graph (Erdős & Rényi, 1959) with the connectivity probability of $p = 0.6$ and the ring graph. For all algorithms, we update the parameters every 4 micro-steps with batch-size $b = 2$. The step sizes of D-SGD, D-SGT and DNSGD are chosen from $\{0.0001, 0.0005, 0.001, 0.005, 0.01, 0.05, 0.1, 0.5\}$. The step sizes of the D-NASA algorithm is given by $\eta_t = m^{1/4}t^{3/4}$ which follows the theory and implementation of Li et al. (2024), where $t$ is the iteration counter and $m$ is the number of agents. For our DNSGD, we set the iterations number of Chebyshev acceleration (Algorithm 1) to be $\hat{K} = K = 2$. We present the numerical results on the Erdös-Rényi graph in Figure 5, and the results on the ring graph are shown in Figure 6. We evaluate the performance of the algorithms by showing the sample complexity and the communication rounds against the loss on the validation set. We can observe that our DNSGD outperforms all baseline methods with respect to both the measures of the computation and the computation.

---

[3]https://github.com/karpathy/nanoGPT

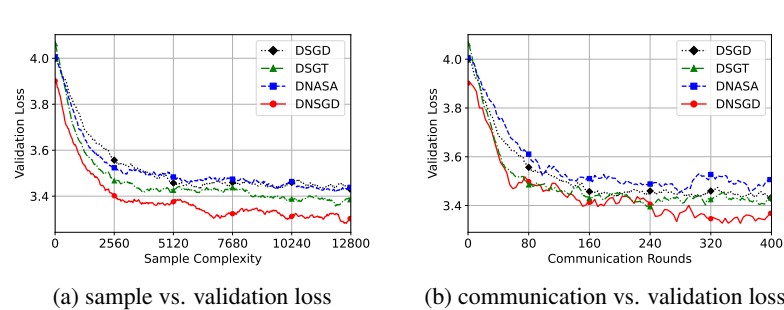

(a) sample vs. validation loss            (b) communication vs. validation loss

Figure 5: We present numerical results for fine-tuning nanoGPT on the shakespeare dataset over the Erdös-Rényi network with $m = 4$ agents. The subfigures (a) and (b) present the sample complexity per agent and the communication rounds against the validation loss.

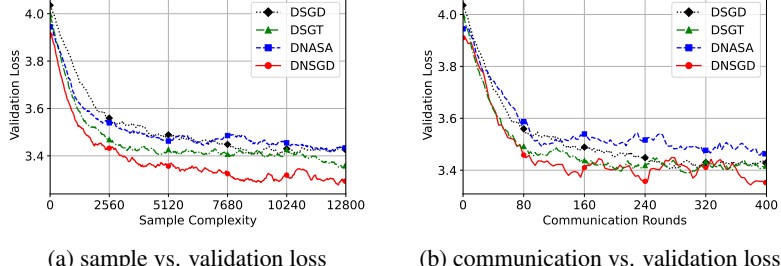

(a) sample vs. validation loss            (b) communication vs. validation loss

Figure 6: We present numerical results for fine-tuning nanoGPT on the shakespeare dataset over the ring network with $m = 4$ agents. The subfigures (a) and (b) present the sample complexity per agent and the communication rounds against the validation loss.