# OpenReview forum: "Decentralized Stochastic Nonconvex Optimization under the Relaxed Smoothness"
_ICLR.cc/2026/Conference — Submitted to ICLR 2026_

### Official Review · Reviewer_TTmR · 2025-10-29

**Soundness:** 4
**Presentation:** 4
**Contribution:** 2
**Rating:** 4
**Confidence:** 3

**Summary:**

This paper studies **decentralized stochastic nonconvex optimization** under *relaxed smoothness* . The authors propose **DNSGD (Decentralized Normalized SGD)**, which integrates normalized gradients, gradient tracking, and Chebyshev-based multi-consensus to address potential unbounded or non-Lipschitz local gradients.
A new **Lyapunov function** involving the product of gradient norm and consensus error is developed to prove convergence to an $\epsilon$-stationary point for each agent.
Theoretical results establish per-agent sample complexity and communication complexity, which reduce to near-optimal known results when $L_1 = 0$.
 Experiments on MNIST and Fashion-MNIST empirically validate the method.

I wish the author could address my concern, and I will improve my score.

**Strengths:**

- The paper is clearly organized, and the main results and proofs are relatively easy to follow.
- Extends decentralized optimization to relaxed-smooth functions.
- When $L_1 = 0$, the obtained convergence rates match the known near-optimal results, which validates the analysis framework.

**Weaknesses:**

- The paper mainly combines *normalized decentralized optimization* with *relaxed smoothness*. Both components have been well studied individually, so the contribution is primarily incremental rather than conceptually new.
- The step size and batch size are quite restrictive — specifically, η = O(ε) and b = O(ε⁻²). These requirements may severely limit practical applicability, and the paper does not provide intuition or empirical justification for them.
- There seem to be mistakes or unclear steps in some proofs, which may affect key results such as the claimed **linear speed-up**. (See Questions 3 and 4 below.)
- Experiments are only performed on MNIST/Fashion-MNIST, which are too simple to demonstrate the benefits of relaxed-smooth analysis.

**Questions:**

**Proposition 1:** Why not directly assume that the global objective $f$ satisfies the relaxed smoothness property?

**Use of normalization:** What is the intrinsic connection between the normalization technique and relaxed smoothness? The normalization does not seem to improve the theoretical complexity, yet it introduces constraints on the batch size.
 I recommend the authors to refer *“Momentum Improves Normalized SGD”*, which uses momentum to alleviate large-batch-size requirements and might provide inspiration here.

**Equation (14):** In the second line, why is it written as an equality? It appears that this should be an inequality, and the scaling term should be $1/m$ instead of $1/m^2$. Please double-check or provide the missing conditions. If my understanding is correct, this issue could invalidate the linear speed-up claim.

**Lemma 5:** How is the first equality derived? It seems the numerator should be $v_i \|\bar{v}\| - \bar{v}\|v_i\|$. Please verify this step carefully.

---

> ### Author Response · Authors · 2025-11-21
>
> Thank you for your careful review and insightful suggestions.
>
> **Q1:** The paper mainly combines normalized decentralized optimization with relaxed smoothness. Both components have been well studied individually, so the contribution is primarily incremental rather than conceptually new.
>
> **A1:** We emphasize that our analysis under the relaxed-smoothness analysis is NOT a simple combination of the known modules.
> As we mentioned in line 187-197, we provide **a new Lyapunov function involving the product of gradient norm and consensus error**, i.e.,
> $$\left(M_0 + M_1||\nabla f({\bar x}^t)||\right)||X^t-1 {\bar x}^t||.$$
> This term align with the measurement unit on the right-hand side of the condition (2) for the relaxed smoothness of the objective,
> which well addresses the consensus errors even if the local gradient is **non-Lipschitz and unbounded**.
>
> In contrast, the existing analysis essentially only considers the **linear combinations** of the function value, gradient norm, and the consensus errors, which requires that the local gradient be Lipschitz continuous (standard smooth) or bounded, so that it does NOT work on the relaxed smoothness (**non-Lipschitz and unbounded local gradient**).
>
> **Q2** The step size and batch size are quite restrictive — specifically, $\eta=O(\epsilon)$ and $b=O(\epsilon^{-2})$. These requirements may severely limit practical applicability, and the paper does not provide intuition or empirical justification for them.
>
> **A2** We emphasize that the settings of $\eta=O(\epsilon)$ and $b=O(\epsilon^{-2})$ are **only for the theoretical analysis**.
> On the other hand, the real-world applications typically do not require $\epsilon$ to be very small,
> so that we do not require the stepsize to be very small or the batch size to be very large in practice.
> For example, we set $b = 250$ and choose $\eta$ from $\\{0.0001,0.0005,0.001,0.005,0.01,0.05,0.1,0.5\\}$ in our experiments (see lines 422-424), which performs well.
>
> Additionally, it is worth noting that the batch size $O(1/\epsilon^2)$ is necessary to simultaneously achieve the tight sample complexity and the tight communication complexity in theoretical (the worst case).
> Recall the lower bounds on the overall sample complexity and the overall communication complexity are $\Omega(1/\epsilon^4)$ and $\Omega(1/\epsilon^2)$, respectively (see lines 239-245 and we focus on the dependency on $\epsilon$ in this response).
> Hence, after every communication round, we need at least the sample complexity of $O(\frac{1/\epsilon^4}{1/\epsilon^2})=O(1/\epsilon^2)$ on average, matching the batch size $O(1/\epsilon^2)$ in our theoretical analysis.
>
> **Q3** Experiments are only performed on MNIST/Fashion-MNIST, which are too simple to demonstrate the benefits of relaxed-smooth analysis.
>
> **A3** Thanks for your suggestion.
> We have provided additional experiments for
> fine-tuning large language models (LLMs) in Appendix I of our rebuttal revision.
> Specifically, we fine-tune nanoGPT ($124$M parameter) on the Shakespeare dataset.
> We perform our experiments on two network topologies with 4 clients, i.e., the Erdős–Rényi graph with the connectivity probability of $p = 0.6$ and the ring graph.
> For all algorithms, we update the parameters every $4$ micro-steps with batch-size $b=2$.
> The step sizes of D-SGD, D-SGT and DNSGD are chosen from $\\{0.0001, 0.0005, 0.001, 0.005, 0.01, 0.05, 0.1, 0.5\\}$.
> The step sizes of the D-NASA algorithm is given by $\eta_t = m^{1/4}t^{3/4}$ which follows the theory and implementation of Li et al. (2024), where $t$ is the iteration counter and $m$ is the number of agents.
>
> In our rebuttal revision (page 23), we present the numerical results on the Erdős–Rényi graph in **Figure 5**, and the results on the ring graph are shown in **Figure 6**.
> We evaluate the performance of the algorithms by showing the sample complexity and the communication rounds against the loss on the validation set.
> We can observe that our DNSGD outperforms all baseline methods with respect to both the measures of computation and the computation.
>
> **Q4** Proposition 1: Why not directly assume that the global objective $f$ satisfies the relaxed smoothness property?
>
> **A4** Only assuming that the global objective satisfies the relaxed smoothness property is not sufficient to guarantee the convergence. To the best of our knowledge, all of existing analysis for decentralized optimization require the assumptions like (relaxed) smoothness or Lipschitz continuous for the local functions.
>
> Taking our analysis as an example, we need to use $||{\bar x}^t - x_i^t||$ to bound $||\nabla f_i({\bar x}^t) - \nabla f_i(x_i^t)||$ in lines 801-805, which depends on the relaxed smoothness of the local function (the relaxed smoothness of the global objective is not enough).

---

> ### Author Response · Authors · 2025-11-21
>
> **Q5** Use of normalization: What is the intrinsic connection between the normalization technique and relaxed smoothness? The normalization does not seem to improve the theoretical complexity, yet it introduces constraints on the batch size. I recommend the authors to refer “Momentum Improves Normalized SGD”, which uses momentum to alleviate large-batch-size requirements and might provide inspiration here.
>
> We clarify that normalization does NOT constraints on the batch size.
> In fact, the normalization is used to achieve the convergence under the relaxed smoothness, while the batch size is used to improve the communication efficiency.
>
> 1. For the relaxed smooth problem, the normalized (stochastic) gradient is more effective than the standard (stochastic) gradient.
> General speaking, the (stochastic) gradient descent without normalization typically need an additional assumption on gradient norms to guarantee the convergence under the relaxed smooth.
> For more details, we recommend the reviewer refer to Sections 4.1 and 4.2 of Zhang et al. (2020b).
>
> 2. It is worth noting that introducing constraints on the batch size do not improve the **sample complexity**, i.e., we can also achieve the sample complexity of $O(\epsilon^{-4})$ by setting $b=1$ and $T=O(\epsilon^{-4})$.
> However, increasing the batch size is essential to improve the **communication complexity** in distributed setting. Please refer to our response in **A2**.
>
> 3. We thanks the reviewer for pointing out the paper ``Momentum Improves Normalized SGD''. Based on the discussion in **A2**, the large batch size is necessary to achieve the tight communication complexity in distributed setting (at least in the worst case).
> We agree that including the momentum is possible to improve the performance of the algorithm in practice, which is highlighted in line 475 in our rebuttal revision.
>
>
>
>
> **Q6** Equation (14): In the second line, why is it written as an equality? It appears that this should be an inequality, and the scaling term should be $1/m$ instead of $1/m^2$. Please double-check or provide the missing conditions. If my understanding is correct, this issue could invalidate the linear speed-up claim.
>
> **A6** We confirm that the scaling $1/m^2$ is correct because this equation hold in expectation. We detailed clarify this step as follows
> $$
>     {\mathbb E}\left\|\left\|\frac{1}{m}\sum_{i=1}^m (g_i^t - \nabla f_i(x_i^t))\right|\right|^2 \\
> =   \frac{1}{m^2}\left({\mathbb E}\left[\sum_{i=1}^m \left\|\left\|g_i^t - \nabla f_i(x_i^t)\right\|\right\|^2\right]  + \sum_{i=1}^m\sum_{j\neq i} {\mathbb E}\left[\langle g_i^t - \nabla f_i(x_i^t), g_j^t - \nabla f_j(x_j^t)\rangle\right]\right) \\
> =  \frac{1}{m^2}\sum_{i=1}^m {\mathbb E}\left\|\left\|g_i^t - \nabla f_i(x_i^t)\right\|\right\|^2,
> $$
> where the second step is based on the fact that $g_i^t$ and $g_j^t$ are independent for $i\neq j$ which implies their covariance ${\mathbb E}[\langle g_i^t - \nabla f_i(x_i^t), g_j^t - \nabla f_i(x_j^t)\rangle]$ is zero.
>
>
> **Q7** Lemma 5: How is the first equality derived? It seems the numerator should be $v_i |\bar{v}| - \bar{v}|v_i|$. Please verify this step carefully.
>
> **A7** We clarify the detailed derivation for this step as follows. Note that we have
> $$
>     \frac{1}{m}\sum_{i=1}^m \frac{v_i}{||v_i||} - \frac{\bar v}{||\bar v||} \\
> =  \frac{1}{m}\sum_{i=1}^m \frac{v_i}{||v_i||} - \frac{1}{m}\sum_{i=1}^m\frac{v_i}{||\bar v||}   \\
> =  \frac{1}{m}\sum_{i=1}^m \frac{(||\bar v||-||v_i||)v_i}{||v_i|| ||\bar v||},
> $$
> where the first step is based on the fact $\bar v=\frac{1}{m}\sum_{i=1}^m v_i$. Taking the norm on above equation leads to the desired equality in the proof of Lemma 5.
> We have incorporated above steps into our rebuttal revision and highlighted it by blue (lines 672-684).
>
> **References**
>
> Jiaxiang Li, Xuxing Chen, Shiqian Ma, and Mingyi Hong. Problem-parameter-free decentralized nonconvex stochastic optimization. arXiv preprint arXiv:2402.08821, 2024.
>
> Jingzhao Zhang, Tianxing He, Suvrit Sra, and Ali Jadbabaie. Why gradient clipping accelerates training: A theoretical justification for adaptivity. In International Conference on Learning Representations, 2020b.

---

### Official Review · Reviewer_x2Bb · 2025-11-02

**Soundness:** 2
**Presentation:** 2
**Contribution:** 2
**Rating:** 4
**Confidence:** 4

**Summary:**

This paper investigates decentralized stochastic nonconvex optimization under the relaxed (L0, L1)-smooth condition. The authors propose Decentralized Normalized Stochastic Gradient Descent (DNSGD), an algorithm that ensures every agent converges to an $\epsilon$-stationary point under (L0, L1) conditions. In the special case of L1 = 0, the above results (nearly) match the lower bounds of decentralized stochastic nonconvex optimization under the standard smoothness.

**Strengths:**

- Develops a new decentralized algorithm that can converge in (L0, L1)-smooth condition.

- Establishes rigorous convergence guarantees for the algorithm.

- Achieves tight bounds when L1 = 0

**Weaknesses:**

1. **Novelty.** I have concerns about the novelty of this work. The paper appears to be a straightforward extension of decentralized gradient tracking under standard smoothness to the more general $(L_0, L_1)$-smooth setting. The main theoretical development combines several well-known techniques—decentralized gradient tracking, Chebyshev acceleration, increasing batch size, and gradient normalization—without introducing sufficiently new insights or algorithmic innovations.


2. **Gradient dissimilarity.** The proposed algorithm employs gradient tracking, which is typically designed to mitigate the impact of gradient dissimilarity across agents. However, the paper still assumes gradient dissimilarity in its theoretical setup, which seems inconsistent with the purpose of gradient tracking. The authors should clarify why this assumption remains necessary.


3. **Experiments.** The experimental validation is relatively weak. Although the paper aims to address $(L_0, L_1)$-smooth problems, the experiments are limited to simple image classification tasks on MNIST and FashionMNIST. These datasets are commonly used in decentralized optimization under standard $L_0$-smoothness and may not adequately reflect the benefits of handling $(L_0, L_1)$-smooth objectives. The authors are encouraged to conduct more **challenging and representative experiments**, such as **LLM pre-training or fine-tuning**, where the loss functions are empirically closer to the $(L_0, L_1)$-smooth regime.

4. **Limited utility.**  The algorithm requires a batch size that scales with $1/\epsilon^2$, which significantly limits its practicality in real-world applications.

**Questions:**

1. **Highlight novelty and challenges.** Clearly emphasize the novelty and specific challenges involved in extending decentralized algorithms to the $(L_0, L_1)$-smooth setting. This will help readers better understand what makes this generalization nontrivial and how it differs from the standard smooth case.

2. **Clarify gradient dissimilarity assumption.** Provide a clear justification for introducing the gradient dissimilarity assumption. For example, the authors could establish a lower bound for decentralized optimization under the $(L_0, L_1)$-smooth condition, explicitly showing that gradient dissimilarity naturally arises and cannot be ignored.

3. **Strengthen experiments.** Include more challenging and representative experiments—such as **LLM pre-training** or **fine-tuning**—to better demonstrate the superiority and practical relevance of the proposed algorithm in the $(L_0, L_1)$-smooth regime.

---

> ### Author Response · Authors · 2025-11-21
>
> Thank you for your insightful review and suggestions.
>
> **Q1**  Highlight novelty and challenges. Clearly emphasize the novelty and specific challenges involved in extending decentralized algorithms to the $(L_0, L_1)$-smooth setting. This will help readers better understand what makes this generalization nontrivial and how it differs from the standard smooth case.
>
> **A1** In lines 187-197, we have emphasized the novelty and specific challenges in extending decentralized algorithms to the $(L_0, L_1)$-smooth setting.
> Specifically, we provide **a new Lyapunov function involving the product of gradient norm and consensus error**, i.e.,
> $$\left(M_0 + M_1||\nabla f({\bar x}^t)||\right)||X^t-1 {\bar x}^t||.$$
> This term align with the measurement unit on the right-hand side of the condition (2) for the relaxed smoothness of the objective,
> which well addresses the consensus errors even if the local gradient is **non-Lipschitz** and **unbounded**.
>
> In contrast, the existing analysis essentially only considers the **linear combinations** of the function value, gradient norm, and the consensus errors, which requires that the local gradient be Lipschitz continuous (standard smooth) or bounded, so that it does NOT work on the relaxed smoothness (**non-Lipschitz** and **unbounded** local gradient).
>
>
> **Q2** Clarify gradient dissimilarity assumption. Provide a clear justification for introducing the gradient dissimilarity assumption. For example, the authors could establish a lower bound for decentralized optimization under the $(L_0, L_1)$-smooth condition, explicitly showing that gradient dissimilarity naturally arises and cannot be ignored.
>
> **A2** It remains an open question whether the assumption of gradient dissimilarity can be removed in decentralized optimization under the relaxed smoothness.
> In line 341-344, we have explained why this assumption is necessary for our convergence analysis, i.e.,
>
> 1.  We use gradient dissimilarity assumption to establish the inequality
> $||\nabla f({\bar x}^{t+1})|| \leq (L_0+L_1\zeta)\eta + \left(1+L_1\eta\right)||\nabla f({\bar x}^{t})||$,
> which connects the gradients at the points ${\bar x}^{t}$ and ${\bar x}^{t+1}$ (see Lemma 9 in Appendix C).
>
> 2. We combine the gradient dissimilarity assumption with the $(L_0,L_1)$-smoothness of local functions (Assumption 2) to guarantee the global objective $f=\frac{1}{m}\sum_{i=1}^m f_i$ is $(L_0+L_1\zeta,L_1)$-smooth (see the proof in Appendix B).
>
> Additionally, the assumption of gradient dissimilarity is commonly used even in **the centralized optimization under the standard smoothness** (Assran et al., 2019; Gorbunov et al., 2021; Lian et al., 2017; Woodworth et al., 2020).
>
>
> **Q3** The proposed algorithm employs gradient tracking, which is typically designed to mitigate the impact of gradient dissimilarity across agents. However, the paper still assumes gradient dissimilarity in its theoretical setup, which seems inconsistent with the purpose of gradient tracking. The authors should clarify why this assumption remains necessary.
>
> **A3** We think the assumption of gradient dissimilarity is NOT inconsistent with the gradient tracking.
>
> 1. As we mentioned in A2, the assumption of gradient dissimilarity is mainly used to address the analysis for the $(L_0,L_1)$-smoothness in our setting, which is not directly related to gradient tracking.
>
> 2. Under the assumption of gradient dissimilarity, the algorithm without gradient tracking can guarantee the convergence for the standard smooth setting, but its communication complexity will include additional dependency on $\log\zeta$, where $\zeta$ is the upper bound of $||\nabla f_i(x) - \nabla f(x)||$ (Yuan et al., 2022, Section 5).
> In other words, employing the gradient tracking can avoid the dependency on $\log\zeta$ in the communication complexity, which is beneficial when $\zeta$ is very large.

---

> ### Author Response · Authors · 2025-11-21
>
> **Q4** Strengthen experiments. Include more challenging and representative experiments—such as LLM pre-training or fine-tuning—to better demonstrate the superiority and practical relevance of the proposed algorithm in the $(L_0, L_1)$-smooth regime.
>
> **A4** Thanks for your suggestion.
>
> We have provided additional experiments for
> fine-tuning large language models (LLMs) in Appendix I of our rebuttal revision.
> Specifically, we fine-tune nanoGPT ($124$M parameter) on the Shakespeare dataset.
> We perform our experiments on two network topologies with 4 clients, i.e., the Erdős–Rényi graph with the connectivity probability of $p = 0.6$ and the ring graph.
> For all algorithms, we update the parameters every $4$ micro-steps with batch-size $b=2$.
> The step sizes of D-SGD, D-SGT and DNSGD are chosen from $\\{0.0001, 0.0005, 0.001, 0.005, 0.01, 0.05, 0.1, 0.5\\}$.
> The step sizes of the D-NASA algorithm is given by $\eta_t = m^{1/4}t^{3/4}$ which follows the theory and implementation of Li et al. (2024), where $t$ is the iteration counter and $m$ is the number of agents.
>
> In our rebuttal revision (page 23), we present the numerical results on the  Erdős–Rényi graph in **Figure 5**, and the results on the ring graph are shown **in Figure 6**.
> We evaluate the performance of the algorithms by showing the sample complexity and the communication rounds against the loss on the validation set.
> We can observe that our DNSGD outperforms all baseline methods with respect to both the measures of computation and the computation.
>
> **Q5** The algorithm requires a batch size that scales with $1/\epsilon^2$, which significantly limits its practicality in real-world applications.
>
> **A5** We emphasize that the batch size $O(1/\epsilon^2)$ is **only for the theoretical analysis**.
> On the other hand, the real-world applications typically do not require $\epsilon$ to be very small,
> so that we do not require the batch size to be very large in practice.
> For example, we set $b = 250$ in our experiments (see line 422), which performs well.
>
> Additionally, it is worth noting that the batch size $O(1/\epsilon^2)$ is necessary to simultaneously achieve the tight sample complexity and the tight communication complexity in theoretical (the worst case).
> Recall the lower bounds on the overall sample complexity and the overall communication complexity are $\Omega(1/\epsilon^4)$ and $\Omega(1/\epsilon^2)$, respectively (see lines 239-245 and we focus on the dependency on $\epsilon$ in this response).
> Hence, after every communication round, we need at least the sample complexity of $O(\frac{1/\epsilon^4}{1/\epsilon^2})=O(1/\epsilon^2)$ on average, matching the batch size $O(1/\epsilon^2)$ in our theoretical analysis.
>
> **References**
>
> Mahmoud Assran, Nicolas Loizou, Nicolas Ballas, and Mike Rabbat. Stochastic gradient push for distributed deep learning. In International Conference on Machine Learning, pp. 344–353, 2019.
>
> Eduard Gorbunov, Filip Hanzely, and Peter Richtárik. Local SGD: Unified theory and new efficient methods. In International Conference on Artificial Intelligence and Statistics, pp. 3556–3564,
> 2021.
>
> Xiangru Lian, Ce Zhang, Huan Zhang, Cho-Jui Hsieh, Wei Zhang, and Ji Liu. Can decentralized algorithms outperform centralized algorithms? a case study for decentralized parallel stochastic gradient descent. In Advances in Neural Information Processing Systems, pp. 5336–5346, 2017.
>
> Blake E. Woodworth, Kumar Kshitij Patel, and Nati Srebro. Minibatch vs local SGD for heterogeneous distributed learning. In Advances in Neural Information Processing Systems, pp. 6281–6292,
> 2020.
>
> Kun Yuan, Xinmeng Huang, Yiming Chen, Xiaohan Zhang, Yingya Zhang, and Pan Pan. Revisiting optimal convergence rate for smooth and non-convex stochastic decentralized optimization. InAdvances in Neural Information Processing Systems, pp. 36382–36395, 2022.
>
> Jiaxiang Li, Xuxing Chen, Shiqian Ma, and Mingyi Hong. Problem-parameter-free decentralized nonconvex stochastic optimization. arXiv preprint arXiv:2402.08821, 2024.

---

### Official Review · Reviewer_BZAg · 2025-11-02

**Soundness:** 3
**Presentation:** 2
**Contribution:** 3
**Rating:** 6
**Confidence:** 3

**Summary:**

This paper studies a distributed optimization problem in which the local objectives satisfy a relaxed smoothness property. The notion of relaxed smoothness was originally introduced by Zhang et al. (2020b), and this work adopts a weaker variant proposed by Zhang et al. (2020a). Under this condition, local gradients are neither Lipschitz continuous nor bounded. The analysis further assumes bounded gradient dissimilarity, which plays a crucial role in establishing the relaxed smoothness of the aggregated objective, as shown in Proposition 3.

While relaxed smoothness has been extensively explored in recent years—particularly in centralized settings for training neural network models—this paper advances the study to decentralized optimization. Notably, the proposed method does not employ gradient clipping. Instead, it integrates normalized gradient descent, gradient tracking, and multi-consensus with Chebyshev acceleration. The authors derive both sample complexity and communication complexity results, showing that their bounds match or surpass the best-known results in certain special cases.

The most relevant prior work is that of Jiang et al. (2025), which also studies first-order decentralized optimization under the relaxed smoothness assumption. In contrast, Jiang et al. employ gradient clipping based on access to the mean vector and exact local gradients. The present paper argues that such an approach is not fully decentralized and stochastic.

**Strengths:**

- This paper extends decentralized nonconvex optimization to the relaxed-smoothness regime, where gradients may be unbounded, which has rarely been studied in decentralized contexts.
- The algorithm presented achieves near-optimal sample and communication complexities.

**Weaknesses:**

- The algorithmic components (normalization, gradient tracking, Chebyshev multi-consensus) are adaptations of known modules; the contribution lies mainly in combining them under the relaxed-smoothness analysis rather than introducing a fundamentally new decentralized mechanism.
- There is no comparison with Jiang et al. (2025) in terms of sample and communication complexities and experimental performance. They can still note that Jiang et al. (2025)'s approach requires additional sampling and communication rounds to ensure a common mean vector and exact local gradient.
- The presentation is highly technical. Motivating problems and a qualitative description of the framework could significantly improve the paper. For example, the paper directly imposes Assumption 5 on the mixing matrix without explicitly describing the underlying communication network.

**Questions:**

How does the presented algorithm perform in comparison to Jiang et al (2025)'s algorithm in terms of sampling and communication complexities and experimental performance?

---

> ### Author Response · Authors · 2025-11-21
>
> Thank you for your careful review and insightful suggestions.
>
> **Q1** The algorithmic components (normalization, gradient tracking, Chebyshev multi-consensus) are adaptations of known modules; the contribution lies mainly in combining them under the relaxed-smoothness analysis rather than introducing a fundamentally new decentralized mechanism.
>
> **A1** We emphasize that our analysis under the relaxed-smoothness analysis is NOT a simple combination of the known modules.
> As we mentioned in line 187-197, we provide **a new Lyapunov function involving the product of gradient norm and consensus error**, i.e.,
> $$\left(M_0 + M_1||\nabla f({\bar x}^t)||\right)||X^t-1 {\bar x}^t||.$$
> This term align with the measurement unit on the right-hand side of the condition (2) for the relaxed smoothness of the objective,
> which well addresses the consensus errors even if the local gradient is **non-Lipschitz** and **unbounded**.
>
> In contrast, the existing analysis essentially only considers the **linear combinations** of the function value, gradient norm, and the consensus errors, which requires that the local gradient be Lipschitz continuous (standard smooth) or bounded, so that it does NOT work on the relaxed smoothness (**non-Lipschitz** and **unbounded** local gradient).
>
>
> **Q2** There is no comparison with Jiang et al. (2025) in terms of sample and communication complexities and experimental performance. They can still note that Jiang et al. (2025)'s approach requires additional sampling and communication rounds to ensure a common mean vector and exact local gradient.
>
> **A2** Note that both Jiang et al. (2025) and our works focus on the **stochastic setting**, i.e., each local function has the **expectation** form
> $$
> f_i(x) = {\mathbb E}[F_i(x;\xi_i)],
> $$
> and we can only access the **stochastic first-order oracle** $\nabla F_i(x;\xi_i)$ by sampling $\xi_i\sim D_i$ for given $x$.
> This implies there is no general way to access the **exact local gradient** $\nabla f_i(x)$ to implement the method of Jiang et al. (2025) in the stochastic setting.
> In other words, their method is not truly decentralized and stochastic  (as stated in lines 330-337), so that we do not compare it with our fully decentralized and stochastic.
>
> Although it is possible to approximate the mean vector and exact local gradient by additional sampling and communication rounds, how to well incorporate the inexact mean vector and local gradient into their framework is unclear.
>
> **Q3** The presentation is highly technical. Motivating problems and a qualitative description of the framework could significantly improve the paper. For example, the paper directly imposes Assumption 5 on the mixing matrix without explicitly describing the underlying communication network.
>
> **A3** Thanks for your suggestion. In our rebuttal revision, we have involved the explanation for Assumption 5 in lines 136-137. Specifically, Assumption 5 implies the mixing matrix $W$ is doubly stochastic, which indicates that multiplying by $W$ on the aggregated matrix encourages all local variables converge to their average.
>
> **Reference**
>
> Zhanhong Jiang, Aditya Balu, and Soumik Sarkar. Decentralized relaxed smooth optimization with gradient descent methods. arXiv preprint arXiv:2508.08413, 2025.

---

### Official Review · Reviewer_jz63 · 2025-11-07

**Soundness:** 3
**Presentation:** 3
**Contribution:** 3
**Rating:** 6
**Confidence:** 4

**Summary:**

This paper proposes a DNSGD method for solving distributed stochastic non-convex problems under the assumption of $(L_0, L_1)$-smoothness. The method is designed by combining normalized gradient descent, gradient tracking, and accelerated gossip procedures. The authors provide theoretical guarantees for the method and compare it to concurrent work. They also provide experiments that validate the theoretical results.

**Strengths:**

1. The proposed results are novel and have theoretical guarantees.

2. The proposed algorithm has significantly better performance compared to concurrent work (see Section 5).

**Weaknesses:**

1. P. 19 line 981. I think the term $\frac{4\eta\rho\sigma}{\sqrt{b}}$ was missed.  Will this affect other estimates, for example, on $b, T$, and $K$ on pages 20 and 21?

**Minor comments:**


1. p.1 lines 41-44. I think the work arXiv:2507.09823 should be cited in the context of recent results for relaxed smoothness.

2. P.15 line 801-802. I think this transition should include a reference to the fact that $||\overline{x}^t - x_i^t|| \leq \frac{1}{L_1}$, because this fact is provided only on page 17 in the proof of another lemma.

3. P. 19 line 1003-1014. Can you explain these transitions carefully? I am confused because on page 17, line 887 there are estimates for $\rho$ and it seems necessary to prove that these estimates are less than the terms in lines 1003-1008.

**Typos:**

1. P.16 line 824. $t+1 \to t$

2. P.16 line 862. “First” $\to$ “second”

3. P.17 line 888. I think some constants have been missed.

4. P.17 line 903. $= \ \to \ \leq$

5. P.17 line 917. Constant 3 has been missed.

6. P.18 line 950. $b \to \sqrt{b}$

**Questions:**

I provided my questions in weaknesses section.

---

> ### Author Response · Authors · 2025-11-21
>
> Thank you for your careful review and insightful suggestions.
>
> **Q1:** P. 19 line 981. I think the term $\frac{4\eta\rho\sigma}{\sqrt{b}}$ was missed. Will this affect other estimates, for example, on $b, T$, and $K$ on pages 20 and 21?
>
> **A1:** Thanks for your careful review. We agree that line 981 (line 1007 in rebuttal version) should contain the term $\frac{4\eta\rho\sigma}{\sqrt{b}}$.
> Addressing this problem only affect the logarithmic term in the setting of $K$, and it does not affect the setting of other parameters such as $b$ and $T$.
> We have fixed this typo by additionally including the blue-highlighted terms on pages 19 and 20 in our rebuttal revision.
> We can see that it only affect the setting of $\rho$, which means we only need to replace
> $K={\mathcal O}\left(\frac{\log m}{\sqrt{\gamma}}\right)$
> with
> $K={\mathcal O}\left(\frac{\log (\sqrt{m}(1+\eta))}{\sqrt{\gamma}}\right)$.
> By using the notation $\tilde{\mathcal O}(\cdot)$ to hide the log factors, we still have $K=\tilde{\mathcal O}\left(\frac{1}{\sqrt{\gamma}}\right)$.
> Hence, fixing this typo does not affect all of our main results.
>
>
>
> **Q2:** p.1 lines 41-44. I think the work arXiv:2507.09823 should be cited in the context of recent results for relaxed smoothness.
>
> **A2:** Thanks for your sharing this work. I have cited this work in lines 40-41 in our rebuttal reversion.
>
>
> **Q3:** P.15 line 801-802. I think this transition should include a reference to the fact that $||\overline{x}^t - x_i^t|| \leq \frac{1}{L_1}$, because this fact is provided only on page 17 in the proof of another lemma.
>
> **A3:** Thanks for your suggestion. We have included a reference for the fact $||\overline{x}^t - x_i^t|| \leq \frac{1}{L_1}$ in our rebuttal revision (lines 811-812).
>
> **Q4:** P. 19 line 1003-1014. Can you explain these transitions carefully? I am confused because on page 17, line 887 there are estimates for $\rho$ and it seems necessary to prove that these estimates are less than the terms in lines 1003-1008.
>
> **A4:** We detailed explain these transitions as follows.
>
> 1. We first explain the conditions in lines 1003-1008.
> Note that the first term in the min operator is $\frac{1}{1+m \eta L_1}$ (lines 1003-1005), matching the estimate in line 887 (page 17).
> In addition, the other five terms in the min operator (lines 1003-1008) corresponds to the five conditions in lines 993-1000.
> Hence, the estimate in lines 993-1000 is less than the estimate in line 887 (page 17).
> In other words, if lines 1003-1008 holds, then line 887 (page 17) naturally holds.
>
> 2. We desire to guarantee the conditions in lines 1003-1008 holds to finish the proof of this theorem.
> Taking the first item in lines 1007-1008 as an example,
> we want to set appropriate $\rho$ to achieve
> $$\rho \leq \dfrac{1}{8(3\sqrt{m}\eta M_1(1 + L_1\eta) + 2\sqrt{m}\eta M_3)}.$$
> The parameter settings implies
> $$
>  \dfrac{1}{8(3\sqrt{m}\eta M_1(1 + L_1\eta) + 2\sqrt{m}\eta M_3)} \geq \Theta\left(\dfrac{1}{8(3\sqrt{m}\eta L_1(1 + 1/2) + 2\sqrt{m}\eta L_1)}\right)
>  \geq \Theta\left(\frac{1}{\sqrt{m}}\right),
> $$
> where the first step uses the facts $M_1=\Theta(L_1)$, $L_1\eta \leq 1/2$, and $M_3\leq \Theta(L_1)$;
> and the second step uses the facts $L_1\eta \leq 1/2$.
> Combining above two inequalities, we know that setting $\rho \leq O(1/\sqrt{m})$ can guarantee the condition
> $$\rho \leq \dfrac{1}{8(3\sqrt{m}\eta M_1(1 + L_1\eta) + 2\sqrt{m}\eta M_3)}.$$
> holds.
> By processing the other terms in the min operator by the similar way, we achieve the requirements of $\rho\leq O(1)$ and $\rho\leq O(1/(\sqrt{m}\eta))$.
> Combining all above requirements of $\rho$,
> we know that taking $\rho \leq O\left(\frac{1}{\sqrt{m}(1+\eta)}\right)$ as lines 1012-1014 guarantees the conditions in lines 1003-1008 hold.
> Consequently, the conditions in lines 993-1000 also hold, which means we can finish the proof.
>
> If the reviewer has any questions about the above response, welcome to the further discussion.
>
>
>
> **Q5:** Typos on:
>
>
> P.16 line 824. $t+1 \to t$
>
> P.16 line 862. “First” $\to$ “second”
>
> P.17 line 888. I think some constants have been missed.
>
> P.17 line 903. $= \ \to \ \leq$
>
> P.17 line 917. Constant 3 has been missed.
>
> P.18 line 950. $b \to \sqrt{b}$
>
> **A5:** Thanks for your careful review. We have fixed these typos in rebuttal revision.
> All of the modifications have been highlighted by blue.
>
> For line 888, we have adjusted the constant for the setting of $K$ in Lemma 3 to guarantee the upper bound of $\rho$ holds.
>
> For line 917, we have added the constant 3 to the expressions of $M_2$ and $M_3$ in line 918.

---

### Author Response · Authors · 2025-11-27

Dear Area Chair and Reviewers,

We sincerely appreciate your taking the time to handle our submission. Based on the comments from the reviewers, we have addressed the concerns and updated our submission accordingly. However, we have not received the response from the reviewers.

We would be very grateful if the reviewers could participate in the discussion. We believe the further response would greatly help us improve our work.

Best regards,

Authors

---

### Author Response · Authors · 2025-11-30
**Rebuttal Summary**

We sincerely appreciate the area chairs and reviewers for taking the time to handle our submission. We thank all the reviewers for their positive feedback on our work. In particular, they noted that our algorithm significantly outperforms concurrent methods; achieves near-optimal sample and communication complexities; attains tight bounds when $L_1 = 0$; is clearly organized; and presents main results and proofs that are relatively easy to follow.

At the rebuttal stage, we have addressed reviewers' comments and incorporated their suggestions into revision highlight the modification in blue). Reviewer TTmR stated "I wish the author could address my concern, and I will improve my score." Due to the discussion period has been cut short, we did not received any feedback on our rebuttal.

We summarize our response to the main concerns as follows.

**1. Reviewer jz63 (Q4) was confused about why the estimate in lines 993–1000 is less than the estimate in line 887 (page 17).**

Note that the first term in the min operator is $\frac{1}{1+m \eta L_1}$ (lines 1003–1005), matching the estimate in line 887 (page 17). In addition, the other five terms in the min operator (lines 1003–1008) correspond to the five conditions in lines 993–1000. Hence, the estimate in lines 993–1000 is less than the estimate in line 887 (page 17). In other words, if lines 1003–1008 holds, then line 887 (page 17) naturally holds.

**2. Reviewers BZAg (Q1) and x2Bb (Q1) did not aware the novelty of our analysis.**

We have clearly mentioned the novelty of our analysis in lines 187–197 of our initial submission (also rebuttal revision).
Specifically, we provide a new Lyapunov function involving **the product of gradient norm and consensus error**, i.e., $$\left(M_0 + M_1||\nabla f({\bar x}^t)||\right)||X^t-1 {\bar x}^t||.$$ This term align with the measurement unit on the right-hand side of the condition (2) for the relaxed smoothness of the objective, which well addresses the consensus errors even if the local gradient is **non-Lipschitz** and **unbounded**.

In contrast, the existing analysis essentially only considers the **linear combinations** of the function value, gradient norm, and the consensus errors, which requires that the local gradient be **Lipschitz continuous** or **bounded**.


**3. Reviewers x2Bb (Q4) and TTmR (Q3) suggested performing more challenging experiments such as LLM pre-training or fine-tuning.**

We have provided additional experiments for fine-tuning nanoGPT on the Shakespeare dataset. The empirical results (Figures 5 and 6 on page 23 in rebuttal revision) show that our DNSGD outperforms all baseline methods with respect to both the measures of computation and communication.

**4. Reviewers x2Bb (Q5) and TTmR (Q2) are not familiar with the setting of batch size $O(1/\epsilon^2)$ in stochastic optimization.**

We emphasized that the batch size $O(1/\epsilon^2)$ is **only for the theoretical analysis**. On the other hand, the real-world applications typically do not require $\epsilon$ to be very small, so that we do not require the batch size to be very large in practice.

Additionally, the batch size $O(1/\epsilon^2)$ is a **standard** setting in the analysis of stochastic distributed optimization, which is **necessary** to simultaneously achieve the tight sample complexity and the tight communication complexity in theoretical.
Recall the lower bounds on the overall sample complexity and the overall communication complexity are $\Omega(1/\epsilon^4)$ and $\Omega(1/\epsilon^2)$, respectively (see lines 239-245 and we focus on the dependency on $\epsilon$ in this response).
Hence, after every communication round, we need at least the sample complexity of $O(\frac{1/\epsilon^4}{1/\epsilon^2})=O(1/\epsilon^2)$ on average, matching the batch size $O(1/\epsilon^2)$ in our theoretical analysis.


**5. Reviewer TTmR (Q6) did not understand the proof of Equation (14).**

We confirm this step is **correct**, i.e., we have $$ {\mathbb E}\left|\left|\frac{1}{m}\sum_{i=1}^m (g_i^t - \nabla f_i(x_i^t))\right|\right|^2 \ = \frac{1}{m^2}\left({\mathbb E}\left[\sum_{i=1}^m \left|\left|g_i^t - \nabla f_i(x_i^t)\right|\right|^2\right] + \sum_{i=1}^m\sum_{j\neq i} {\mathbb E}\left[\langle g_i^t - \nabla f_i(x_i^t), g_j^t - \nabla f_j(x_j^t)\rangle\right]\right) \ = \frac{1}{m^2}\sum_{i=1}^m {\mathbb E}\left|\left|g_i^t - \nabla f_i(x_i^t)\right|\right|^2,$$
where the second step is based on $g_i^t$ and $g_j^t$ are **independent** for $i\neq j$ which implies their covariance is zero.

**6. Reviewer TTmR (Q7) did not understand the first equality in derivation of Lemma 5.**

We confirm this step is **correct**, i.e., we have $$ \frac{1}{m}\sum_{i=1}^m\frac{v_i}{||v_i||}-\frac{\bar v}{||\bar v||}\=\frac{1}{m}\sum_{i=1}^m\frac{v_i}{||v_i||}-\frac{1}{m}\sum_{i=1}^m\frac{v_i}{||\bar v||}\=\frac{1}{m}\sum_{i=1}^m\frac{(||\bar v||-||v_i||)v_i}{||v_i|| ||\bar v||},$$ where the first step is based on $\bar v=\frac{1}{m}\sum_{i=1}^m v_i$.

---

### Meta-Review · Area_Chair_7DCd · 2025-12-22

**Summary:**

This paper studies decentralized stochastic nonconvex optimization under the (L0,L1)-smoothness assumption. Sample complexity is obtained for the proposed algorithm. Based on the reviews and the AC’s own reading, the contribution of this paper appears to be incremental. Specifically, the motivation of studying the problem under the relaxed smoothness assumption is not clearly discussed, it is thus not clear where the algorithm can be applied. The novelty is limited because the algorithm and analysis are straightforward extension of decentralized gradient tracking algorithm. The presence of the gradient dissimilarity assumption is also a drawback because in the regular smoothness setting this assumption can be removed when gradient tracking is employed. The author’s rebuttal on these issues are not satisfying.

**Reviewer Concerns:**

Addressed: Clarification of some parts of the proofs.
Outstanding: motivation, novelty, usage.

**Reviewer Scores:**

This paper lacks a clear motivation, requires strong assumption, and the contribution is incremental. Reviewers who raised these questions, such as Reviewers BZAg, x2Bb, TTmR are not likely to increase their scores.

---

### Decision · Program_Chairs · 2026-01-26

Reject